# Torsional Diffusion for Molecular Conformer Generation

**Bowen Jing,**[*][1] **Gabriele Corso,**[*][1] **Jeffrey Chang,**[2] **Regina Barzilay,**[1] **Tommi Jaakkola**[1]
[1]CSAIL, Massachusetts Institute of Technology   [2]Dept. of Physics, Harvard University

## Abstract

Molecular conformer generation is a fundamental task in computational chemistry. Several machine learning approaches have been developed, but none have outperformed state-of-the-art cheminformatics methods. We propose *torsional diffusion*, a novel diffusion framework that operates on the space of torsion angles via a diffusion process on the hypertorus and an extrinsic-to-intrinsic score model. On a standard benchmark of drug-like molecules, torsional diffusion generates superior conformer ensembles compared to machine learning and cheminformatics methods in terms of both RMSD and chemical properties, and is orders of magnitude faster than previous diffusion-based models. Moreover, our model provides exact likelihoods, which we employ to build the first generalizable Boltzmann generator. Code is available at `https://github.com/gcorso/torsional-diffusion`.

## 1 Introduction

Many properties of a molecule are determined by the set of low-energy structures, called *conformers*, that it adopts in 3D space. Conformer generation is therefore a fundamental problem in computational chemistry [Hawkins, 2017] and an area of increasing attention in machine learning. Traditional approaches to conformer generation consist of metadynamics-based methods, which are accurate but slow [Pracht et al., 2020]; and cheminformatics-based methods, which are fast but less accurate [Hawkins et al., 2010, Riniker and Landrum, 2015]. Thus, there is growing interest in developing deep generative models to combine high accuracy with fast sampling.

Diffusion or score-based generative models [Ho et al., 2020, Song et al., 2021]—a promising class of generative models—have been applied to conformer generation under several different formulations. These have so far considered diffusion processes in *Euclidean* space, in which Gaussian noise is injected independently into every data coordinate—either pairwise distances in a distance matrix [Shi et al., 2021, Luo et al., 2021] or atomic coordinates in 3D [Xu et al., 2022]. However, these models require a large number of denoising steps and have so far failed to outperform the best cheminformatics methods.

We instead propose *torsional diffusion*, in which the diffusion process over conformers acts only on the torsion angles and leaves the other degrees of freedom fixed. This is possible and effective because the flexibility of a molecule, and thus the difficulty of conformer generation, lies largely in torsional degrees of freedom [Axelrod and Gómez-Bombarelli, 2022]; in particular, bond lengths and angles can already be determined quickly and accurately by standard cheminformatics methods. Leveraging this insight significantly reduces the dimensionality of the sample space; drug-like molecules[2] have, on average, $n = 44$ atoms, corresponding to a $3n$-dimensional Euclidean space, but only $m = 7.9$ torsion angles of rotatable bonds.

---

[*]Equal contribution. Correspondence to `{bjing, gcorso}@mit.edu`.
[2]As measured from the standard dataset GEOM-DRUGS [Axelrod and Gómez-Bombarelli, 2022]

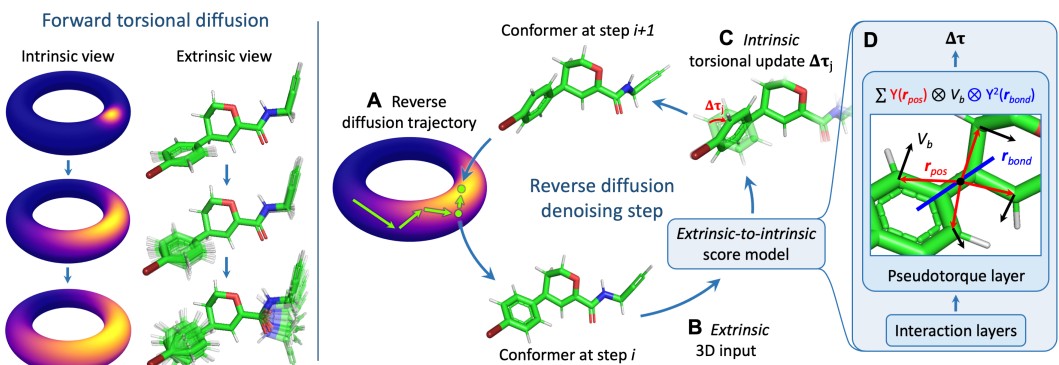

Figure 1: **Overview of torsional diffusion.** *Left*: Extrinsic and intrinsic views of torsional diffusion (only 2 dimensions/bonds shown). *Right*: In a step of reverse diffusion (**A**), the current conformer is provided as a 3D structure (**B**) to the score model, which predicts intrinsic torsional updates (**C**). The final layer of the score model is constructed to resemble a torque computation around each bond (**D**). $Y$ refers to the spherical harmonics and $V_b$ the learned atomic embeddings.

Torsion angle coordinates define not a Euclidean space, but rather an $m$-dimensional torus $\mathbb{T}^m$ (Figure 1, *left*). However, the dimensionality and distribution over the torus vary between molecules and even between different ways of defining the torsional space for the same molecule. To resolve these difficulties, we develop an *extrinsic-to-intrinsic* score model (Figure 1, *right*) that takes as input a 3D point cloud representation of the conformer in Euclidean space (extrinsic coordinates), and predicts as output a score on a torsional space *specific to that molecule* (intrinsic coordinates). To do so, we consider a *torsional score* for a bond as a geometric property of a 3D point cloud, and use $SE(3)$-equivariant networks to predict them directly for each bond.

Unlike prior work, our model provides exact likelihoods of generated conformers, enabling training with the ground-truth *energy* function rather than samples alone. This connects with the literature on *Boltzmann generators*—generative models which aim to sample the Boltzmann distribution of physical systems without expensive molecular dynamics or MCMC simulations [Noé et al., 2019, Köhler et al., 2021]. Thus, as a variation on the torsional diffusion framework, we develop *torsional Boltzmann generators* that can approximately sample the conditional Boltzmann distribution for unseen molecules. This starkly contrasts with existing Boltzmann generators, which are specific for the chemical system on which they are trained.

Our main contributions are:

- We formulate conformer generation in terms of diffusion modeling on the hypertorus— the first demonstration of non-Euclidean diffusion on complex datasets—and develop an extrinsic-to-intrinsic score model that satisfies the required symmetries: $SE(3)$ invariance, torsion definition invariance, and parity equivariance.

- We obtain state-of-the-art results on the GEOM-DRUGS dataset [Axelrod and Gómez-Bombarelli, 2022] and are the first method to consistently outperform the established commercial software OMEGA [Hawkins, 2017]. We do so using two orders of magnitude *fewer* denoising steps than GeoDiff [Xu et al., 2022], the best Euclidean diffusion approach.

- We propose torsional Boltzmann generators—the first Boltzmann generator based on diffusion models rather than normalizing flows and the first to be useful for a class of molecules rather than a specific system.

## 2 Background

**Diffusion generative models** Consider the data distribution as the starting distribution $p_0(\mathbf{x})$ of a *forward diffusion process* described by an Ito stochastic differential equation (SDE):

$$d\mathbf{x} = \mathbf{f}(\mathbf{x}, t)\, dt + g(t)\, d\mathbf{w}, \quad t \in (0, T) \tag{1}$$

where $\mathbf{w}$ is the Wiener process and $\mathbf{f}(\mathbf{x}, t), g(t)$ are chosen functions. With sufficiently large $T$, the distribution $p_T(\mathbf{x})$—the *prior*—approaches a simple Gaussian. Sampling from the prior and solving the *reverse diffusion*

$$d\mathbf{x} = \left[\mathbf{f}(\mathbf{x}_t, t) - g^2(t)\nabla_{\mathbf{x}} \log p_t(\mathbf{x})\right] \, dt + g(t) \, d\bar{\mathbf{w}} \qquad (2)$$

yields samples from the data distribution $p_0(\mathbf{x})$ [Anderson, 1982, Song et al., 2021]. Diffusion, or score-based, generative models [Ho et al., 2020, Song et al., 2021] learn the score $\nabla_{\mathbf{x}} \log p_t(\mathbf{x})$ of the diffused data with a neural network and generate data by approximately solving the reverse diffusion. The score of the diffused data also defines a *probability flow ODE*—a continuous normalizing flow that deterministically transforms the prior into the data distribution [Song et al., 2021]. We leverage the insight that, in many cases, this flow makes it possible to use diffusion models in place of normalizing flows and highlight one such case with the torsional Boltzmann generator.

Diffusion generative models have traditionally been used to model data on Euclidean spaces (such as images); however, De Bortoli et al. [2022] recently showed that the theoretical framework holds with relatively few modifications for data distributions on compact Riemannian manifolds. The hypertorus $\mathbb{T}^m$, which we use to define torsional diffusion, is a specific case of such a manifold.

Several methods [Salimans and Ho, 2022, Vahdat et al., 2021, Nichol and Dhariwal, 2021] have been proposed to improve and accelerate diffusion models in the domain of image generation. Among these, the most relevant to this work is *subspace diffusion* [Jing et al., 2022], in which the diffusion is progressively restricted to linear subspaces. Torsional diffusion can be viewed in a similar spirit, as it effectively restricts Euclidean diffusion to a nonlinear *manifold* given by fixing the non-torsional degrees of freedom.

**Molecular conformer generation** The *conformers* of a molecule are the set of its energetically favorable 3D structures, corresponding to local minima of the potential energy surface.[3] The gold standards for conformer generation are metadynamics-based methods such as CREST [Pracht et al., 2020], which explore the potential energy surface while filling in local minima [Hawkins, 2017]. However, these require an average of 90 core-hours per drug-like molecule [Axelrod and Gómez-Bombarelli, 2022] and are not considered suitable for high-throughput applications. Cheminformatics methods instead leverage approximations from chemical heuristics, rules, and databases for significantly faster generation [Lagorce et al., 2009, Cole et al., 2018, Miteva et al., 2010, Bolton et al., 2011, Li et al., 2007]; while these can readily model highly constrained degrees of freedom, they fail to capture the full energy landscape. The most well-regarded of such methods include the commercial software OMEGA [Hawkins et al., 2010] and the open-source RDKit ETKDG [Landrum et al., 2013, Riniker and Landrum, 2015].

A number of machine learning methods for conformer generation has been developed [Xu et al., 2021a,b, Shi et al., 2021, Luo et al., 2021], the most recent and advanced of which are GeoMol [Ganea et al., 2021] and GeoDiff [Xu et al., 2022]. GeoDiff is a Euclidean diffusion model that treats conformers as point clouds $\mathbf{x} \in \mathbb{R}^{3n}$ and learns an $SE(3)$ equivariant score. On the other hand, GeoMol employs a graph neural network that, in a single forward pass, predicts neighboring atomic coordinates and torsion angles from a stochastic seed.

**Boltzmann generators** An important problem in physics and chemistry is that of generating independent samples from a Boltzmann distribution $p(\mathbf{x}) \propto e^{-E(\mathbf{x})/kT}$ with known but unnormalized density.[4] Generative models with exact likelihoods, such as normalizing flows, can be trained to match such densities [Noé et al., 2019] and thus provide independent samples from an approximation of the target distribution. Such *Boltzmann generators* have shown high fidelity on small organic molecules [Köhler et al., 2021] and utility on systems as large as proteins [Noé et al., 2019]. However, a separate model has to be trained for every molecule, as the normalizing flows operate on intrinsic coordinates whose definitions are specific to that molecule. This limits the utility of existing Boltzmann generators for molecular screening applications.

---

[3]Conformers are typically considered up to an energy cutoff above the global minimum.

[4]This is related to but distinct from conformer generation, as conformers are the local minima of the Boltzmann distribution rather than independent samples.

## 3 Torsional Diffusion

Consider a molecule as a graph $G = (\mathcal{V}, \mathcal{E})$ with atoms $v \in \mathcal{V}$ and bonds $e \in \mathcal{E}$,[5] and denote the space of its possible conformers $\mathcal{C}_G$. A conformer $C \in \mathcal{C}_G$ can be specified in terms of its *intrinsic* (or internal) coordinates: local structures $L$ consisting of bond lengths, bond angles, and cycle conformations; and torsion angles $\boldsymbol{\tau}$ consisting of dihedral angles around freely rotatable bonds (precise definitions in Appendix A). We consider a bond *freely rotatable* if severing the bond creates two connected components of $G$, each of which has at least two atoms.[6] Thus, torsion angles in cycles (or rings), which cannot be rotated independently, are considered part of the local structure $L$.

*Conformer generation* consists of learning probability distributions $p_G(L, \boldsymbol{\tau})$. However, the set of possible stable local structures $L$ for a particular molecule is very constrained and can be accurately predicted by fast cheminformatics methods, such as RDKit ETKDG [Riniker and Landrum, 2015] (see Appendix F.1 for verification). Thus, we use RDKit to provide approximate samples from $p_G(L)$, and develop a diffusion-based generative model to learn distributions $p_G(\boldsymbol{\tau} \mid L)$ over torsion angles—conditioned on a given graph and local structure.

Our method is illustrated in Figure 1 and detailed as follows. Section 3.1 formulates diffusion modeling on the torus defined by torsion angles. Section 3.2 describes the torsional score framework, Section 3.3 the required symmetries, and Section 3.4 our score model architecture. Section 3.5 discusses likelihoods, and Section 3.6 how likelihoods can be used for energy-based training.

### 3.1 Diffusion modeling on $\mathbb{T}^m$

Since each torsion angle coordinate lies in $[0, 2\pi)$, the $m$ torsion angles of a conformer define a hypertorus $\mathbb{T}^m$. To learn a generative model over this space, we apply the continuous score-based framework of Song et al. [2021], which holds with minor modifications for data distributions on compact Riemannian manifolds (such as $\mathbb{T}^m$) [De Bortoli et al., 2022]. Specifically, for Riemannian manifold $M$ let $\mathbf{x} \in M$, let $\mathbf{w}$ be the Brownian motion on the manifold, and let the drift $\mathbf{f}(\mathbf{x}, t)$, score $\nabla_{\mathbf{x}} \log p_t(\mathbf{x})$, and score model output $\mathbf{s}(\mathbf{x}, t)$ be elements of the tangent space $T_{\mathbf{x}} M$. Then equation 2 remains valid—that is, discretizing and solving the reverse SDE on the manifold as a *geodesic random walk* starting with samples from $p_T(\mathbf{x})$ approximately recovers the original data distribution $p_0(\mathbf{x})$ [De Bortoli et al., 2022].

For the forward diffusion we use rescaled Brownian motion given by $\mathbf{f}(\mathbf{x}, t) = 0, g(t) = \sqrt{\frac{d}{dt}\sigma^2(t)}$ where $\sigma(t)$ is the noise scale. Specifically, we use an exponential diffusion $\sigma(t) = \sigma_{\min}^{1-t}\sigma_{\max}^{t}$ as in Song and Ermon [2019], with $\sigma_{\min} = 0.01\pi$, $\sigma_{\max} = \pi, t \in (0, 1)$. Due to the compactness of the manifold, however, the prior $p_T(\mathbf{x})$ is no longer a Gaussian, but a *uniform* distribution over $M$.

Training the score model with denoising score matching requires a procedure to sample from the perturbation kernel $p_{t|0}(\mathbf{x}' \mid \mathbf{x})$ of the forward diffusion and compute its score. We view the torus $\mathbb{T}^m \cong [0, 2\pi)^m$ as the quotient space $\mathbb{R}^m / 2\pi\mathbb{Z}^m$ with equivalence relations $(\tau_1, \ldots \tau_m) \sim (\tau_1 + 2\pi, \ldots, \tau_m) \ldots \sim (\tau_1, \ldots \tau_m + 2\pi)$. Hence, the perturbation kernel for rescaled Brownian motion on $\mathbb{T}^m$ is the *wrapped normal distribution* on $\mathbb{R}^m$; that is, for any $\boldsymbol{\tau}, \boldsymbol{\tau}' \in [0, 2\pi)^m$, we have

$$p_{t|0}(\boldsymbol{\tau}' \mid \boldsymbol{\tau}) \propto \sum_{\mathbf{d} \in \mathbb{Z}^m} \exp\left(-\frac{||\boldsymbol{\tau} - \boldsymbol{\tau}' + 2\pi\mathbf{d}||^2}{2\sigma^2(t)}\right) \tag{3}$$

where $\sigma(t)$ is the noise scale of the perturbation kernel $p_{t|0}$. We thus sample from the perturbation kernel by sampling from the corresponding unwrapped isotropic normal and taking elementwise mod $2\pi$. The scores of the kernel are pre-computed using a numerical approximation. During training, we sample times $t$ at uniform and minimize the denoising score matching loss

$$J_{\text{DSM}}(\theta) = \mathbb{E}_t \left[\lambda(t)\mathbb{E}_{\boldsymbol{\tau}_0 \sim p_0, \boldsymbol{\tau}_t \sim p_{t|0}(\cdot|\boldsymbol{\tau}_0)} \left[||\mathbf{s}(\boldsymbol{\tau}_t, t) - \nabla_{\boldsymbol{\tau}_t} \log p_{t|0}(\boldsymbol{\tau}_t \mid \boldsymbol{\tau}_0)||^2\right]\right] \tag{4}$$

where the weight factors $\lambda(t) = 1/\mathbb{E}_{\boldsymbol{\tau} \sim p_{t|0}(\cdot|0)} \left[||\nabla_{\boldsymbol{\tau}} \log p_{t|0}(\boldsymbol{\tau} \mid \mathbf{0})||^2\right]$ are also precomputed. As the tangent space $T_{\boldsymbol{\tau}}\mathbb{T}^m$ is just $\mathbb{R}^m$, all the operations in the loss computation are the familiar ones.

---

[5]Chirality and other forms of stereoisomerism are discussed in Appendix F.3.

[6]Notably, this counts double bonds as rotatable. See Appendix F.3 for further discussion.

For inference, we first sample from a uniform prior over the torus. We then discretize and solve the reverse diffusion with a geodesic random walk; however, since the exponential map on the torus (viewed as a quotient space) is just $\exp_{\boldsymbol{\tau}}(\boldsymbol{\delta}) = \boldsymbol{\tau} + \boldsymbol{\delta} \mod 2\pi$, the geodesic random walk is equivalent to the wrapping of the random walk on $\mathbb{R}^m$.

## 3.2 Torsional score framework

While we have defined the diffusion process over intrinsic coordinates, learning a score model $\mathbf{s}(\boldsymbol{\tau}, t)$ directly over intrinsic coordinates is potentially problematic for several reasons. First, the dimensionality $m$ of the torsional space depends on the molecular graph $G$. Second, the mapping from torsional space to physically distinct conformers depends on $G$ and local structures $L$, but it is unclear how to best provide these to a model over $\mathbb{T}^m$. Third, there is no canonical choice of independent intrinsic coordinates $(L, \boldsymbol{\tau})$; in particular, the torsion angle at a rotatable bond can be defined as any of the dihedral angles at that bond, depending on an arbitrary choice of reference neighbors (Figure 2 and Appendix A). Thus, even with fixed $G$ and $L$, the mapping from $\mathbb{T}^m$ to conformers is ill-defined. This posed a significant challenge to prior works using intrinsic coordinates [Ganea et al., 2021].

To circumvent these difficulties, we instead consider a conformer $C \in \mathcal{C}_G$ in terms of its *extrinsic* (or Cartesian) coordinates—that is, as a point cloud in 3D space, defined up to global roto-translation: $\mathcal{C}_G \cong \mathbb{R}^{3n}/SE(3)$. Then, we construct the score model $\mathbf{s}_G(C, t)$ as a function over $\mathcal{C}_G$ rather than $\mathbb{T}^m$. The outputs remain in the tangent space of $\mathbb{T}^m$, which is just $\mathbb{R}^m$. Such a score model is simply an $SE(3)$-*invariant* model over point clouds in 3D space $\mathbf{s}_G : \mathbb{R}^{3n} \times [0, T] \mapsto \mathbb{R}^m$ conditioned on $G$. Thus, we have reduced the problem of learning a score on the torus, conditioned on the molecular graph and local structure, to the much more familiar problem of predicting $SE(3)$-invariant scalar quantities—one for each bond—from a 3D conformer.

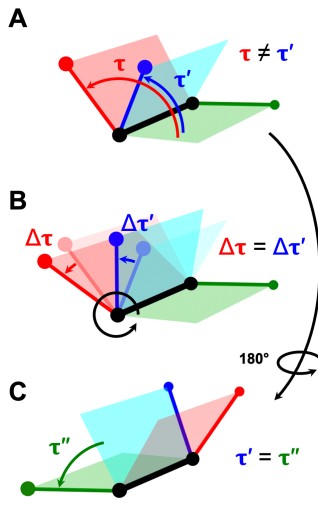

Figure 2: **A**: The torsion $\tau$ around a bond depends on a choice of neighbors. **B**: The *change* $\Delta\tau$ caused by a relative rotation is the same for all choices. **C**: The sign of $\Delta\tau$ is unambiguous because given the same neighbors, $\tau$ does not depend on bond direction.

It may appear that we still need to choose a definition of each torsion angle $\tau_i$ so that we can sample from $p_{t|0}(\cdot \mid \boldsymbol{\tau})$ during training and solve the reverse SDE over $\boldsymbol{\tau}$ during inference. However, we leverage the following insight: given *fixed local structures*, the action on $C$ of changing a single torsion angle $\tau_i$ by some $\Delta\tau_i$ can be applied without choosing a definition (Figure 2). Geometrically, this action is a (signed) relative rotation of the atoms on opposite sides of the bond and can be applied directly to the atomic coordinates in 3D. The geometric intuition can be stated as follows (proven in Appendix B and discussed further in Appendix F.2).

**Proposition 1.** *Let $(b_i, c_i)$ be a rotatable bond, let $\mathbf{x}_{\mathcal{V}(b_i)}$ be the positions of atoms on the $b_i$ side of the molecule, and let $R(\boldsymbol{\theta}, x_{c_i}) \in SE(3)$ be the rotation by Euler vector $\boldsymbol{\theta}$ about $x_{c_i}$. Then for $C, C' \in \mathcal{C}_G$, if $\tau_i$ is any definition of the torsion angle around bond $(b_i, c_i)$,*

$$
\begin{array}{cccc}
\tau_i(C') = \tau_i(C) + \theta & & & \mathbf{x}'_{\mathcal{V}(b_i)} = \mathbf{x}_{\mathcal{V}(b_i)} \\
\tau_j(C') = \tau_j(C) \quad \forall j \neq i & \text{if} & \exists \mathbf{x} \in C, \mathbf{x}' \in C'. & \mathbf{x}'_{\mathcal{V}(c_i)} = R(\theta\, \hat{\mathbf{r}}_{b_i c_i}, x_{c_i})\, \mathbf{x}_{\mathcal{V}(c_i)}
\end{array} \tag{5}
$$

*where $\hat{\mathbf{r}}_{b_i c_i} = (x_{c_i} - x_{b_i})/\|x_{c_i} - x_{b_i}\|$.*

To apply a torsion update $\Delta\boldsymbol{\tau} = (\Delta\tau_1, \ldots \Delta\tau_m)$ involving all bonds, we apply $\Delta\tau_i$ sequentially in any order. Then, since training and inference only make use of torsion updates $\Delta\boldsymbol{\tau}$, we work solely in terms of 3D point clouds and updates applied to them. To draw local structures $L$ from RDKit, we draw full 3D conformers $C \in \mathcal{C}_G$ and then randomize all torsion angles to sample uniformly over $\mathbb{T}^m$. To solve the reverse SDE, we repeatedly predict torsion updates directly from, and apply them directly to, the 3D point cloud. Therefore, since our method never requires a choice of reference neighbors for any $\tau_i$, it is manifestly invariant to such a choice. These procedures are detailed in Appendix C.

### 3.3 Parity equivariance

The torsional score framework presented thus far requires an $SE(3)$-invariant model. However, an additional symmetry requirement arises from the fact that the underlying physical energy is invariant, or extremely nearly so, under *parity inversion* [Quack, 2002]. Thus our learned density should respect $p(C) = p(-C)$ where $-C = \{-\mathbf{x} \mid \mathbf{x} \in C\}$. In terms of the conditional distribution over torsion angles, we require $p(\boldsymbol{\tau}(C) \mid L(C)) = p(\boldsymbol{\tau}(-C) \mid L(-C))$. Then,

**Proposition 2.** *If* $p(\boldsymbol{\tau}(C) \mid L(C)) = p(\boldsymbol{\tau}(-C) \mid L(-C))$*, then for all diffusion times* $t$,

$$\nabla_{\boldsymbol{\tau}} \log p_t(\boldsymbol{\tau}(C) \mid L(C)) = -\nabla_{\boldsymbol{\tau}} \log p_t(\boldsymbol{\tau}(-C) \mid L(-C)) \tag{6}$$

Because the score model seeks to learn $\mathbf{s}_G(C,t) = \nabla_{\boldsymbol{\tau}} \log p_t(\boldsymbol{\tau}(C) \mid L(C))$, we must have $\mathbf{s}_G(C,t) = -\mathbf{s}_G(-C,t)$. Thus, the score model must be *invariant* under $SE(3)$ but *equivariant* (change sign) under parity inversion of the input point cloud— i.e. it must output a set of *pseudoscalars* in $\mathbb{R}^m$.

### 3.4 Score network architecture

Based on sections 3.2 and 3.3, the desiderata for the score model are:

*Predict a pseudoscalar* $\delta\tau_i := \partial \log p/\partial\tau_i \in \mathbb{R}$ *that is* $SE(3)$-*invariant and parity equivariant for every rotatable bond in a 3D point cloud representation of a conformer.*

While there exist several GNN architectures which are $SE(3)$-equivariant [Jing et al., 2021, Satorras et al., 2021], their $SE(3)$-invariant outputs are also parity invariant and, therefore, cannot satisfy the desired symmetry. Instead, we leverage the ability of equivariant networks based on tensor products [Thomas et al., 2018, Geiger et al., 2022] to produce pseudoscalar outputs.

Our architecture, detailed in Appendix D, consists of an embedding layer, a series of atomic convolution layers, and a final bond convolution layer. The first two closely follow the architecture of Tensor Field Networks [Thomas et al., 2018], and produce learned feature vectors for each atom. The final bond convolution layer constructs tensor product filters spatially centered on every rotatable bond and aggregates messages from neighboring atom features. We extract the pseudoscalar outputs of this filter to produce a single real-valued pseudoscalar prediction $\delta\tau_i$ for each rotatable bond.

Naively, the bond convolution layer could be constructed the same way as the atomic convolution layers, i.e., with spherical harmonic filters. However, to supply information about the orientation of the bond about which the torsion occurs, we construct a filter from the product of the spherical harmonics with a representation of the bond (Figure 1D). Because the convolution conceptually resembles computing the torque, we call this final layer the *pseudotorque* layer.

### 3.5 Likelihood

By using the probability flow ODE, we can compute the likelihood of any sample $\boldsymbol{\tau}$ as follows [Song et al., 2021, De Bortoli et al., 2022]:

$$\log p_0(\boldsymbol{\tau}_0) = \log p_T(\boldsymbol{\tau}_T) - \frac{1}{2} \int_0^T g^2(t) \, \nabla_{\boldsymbol{\tau}} \cdot \mathbf{s}_G(\boldsymbol{\tau}_t, t) \, dt \tag{7}$$

In Song et al. [2021], the divergence term is approximated via Hutchinson's method [Hutchinson, 1989], which gives an unbiased estimate of $\log p_0(\boldsymbol{\tau})$. However, this gives a *biased* estimate of $p_0(\boldsymbol{\tau})$, which is unsuitable for our applications. Thus, we compute the divergence term directly, which is feasible here (unlike in Euclidean diffusion) due to the reduced dimensionality of the torsional space.

The above likelihood is in *torsional* space $p_G(\boldsymbol{\tau} \mid L), \boldsymbol{\tau} \in \mathbb{T}^m$, but to enable compatibility with the Boltzmann measure $e^{-E(\mathbf{x})/kT}$, it is desirable to interconvert this with a likelihood in *Euclidean* space $p(\mathbf{x} \mid L), \mathbf{x} \in \mathbb{R}^{3n}$. A factor is necessary to convert between the volume element in torsional space and in Euclidean space (full derivation in Appendix B):

**Proposition 3.** *Let* $\mathbf{x} \in C(\boldsymbol{\tau}, L)$ *be a centered[7] conformer in Euclidean space. Then,*

$$p_G(\mathbf{x} \mid L) = \frac{p_G(\boldsymbol{\tau} \mid L)}{8\pi^2 \sqrt{\det g}} \quad \text{where} \quad g_{\alpha\beta} = \sum_{k=1}^n J_\alpha^{(k)} \cdot J_\beta^{(k)} \tag{8}$$

---

[7]Additional formalism is needed for translations, but it is independent of the conformer and can be ignored.

*where the indices $\alpha, \beta$ are integers between 1 and $m + 3$. For $1 \leq \alpha \leq m$, $J_\alpha^{(k)}$ is defined as*

$$J_i^{(k)} = \tilde{J}_i^{(k)} - \frac{1}{n} \sum_{\ell=1}^{n} \tilde{J}_i^{(\ell)} \quad \text{with} \quad \tilde{J}_i^{(\ell)} = \begin{cases} 0 & \ell \in \mathcal{V}(b_i), \\ \frac{\mathbf{x}_{b_i} - \mathbf{x}_{c_i}}{||\mathbf{x}_{b_i} - \mathbf{x}_{c_i}||} \times (\mathbf{x}_\ell - \mathbf{x}_{c_i}), & \ell \in \mathcal{V}(c_i), \end{cases} \tag{9}$$

*and for $\alpha \in \{m + 1, m + 2, m + 3\}$ as*

$$J_{m+1}^{(k)} = \mathbf{x}_k \times \hat{x}, \qquad J_{m+2}^{(k)} = \mathbf{x}_k \times \hat{y}, \qquad J_{m+3}^{(k)} = \mathbf{x}_k \times \hat{z}, \tag{10}$$

*where $(b_i, c_i)$ is the freely rotatable bond for torsion angle $i$, $\mathcal{V}(b_i)$ is the set of all nodes on the same side of the bond as $b_i$, and $\hat{x}, \hat{y}, \hat{z}$ are the unit vectors in the respective directions.*

### 3.6 Energy-based training

By computing likelihoods, we can train torsional diffusion models to match the Boltzmann distribution over torsion angles using the energy function. At a high level, we minimize the usual score matching loss, but with simulated samples from the Boltzmann distribution rather than data samples. The procedure therefore consists of two stages: resampling and score matching, which are tightly coupled during training (Algorithm 1). In the *resampling* stage, we use the model as an importance sampler for the Boltzmann distribution, where Proposition 3 is used to compute the (unnormalized) torsional Boltzmann density $\tilde{p}_G(\boldsymbol{\tau} \mid L)$. In the *score-matching* stage, the importance weights are used to approximate the denoising score-matching loss with expectations taken over $\tilde{p}_G(\boldsymbol{\tau} \mid L)$. As the model learns the score, it improves as an importance sampler.

This training procedure differs substantially from that of existing Boltzmann generators, which are trained as flows with a loss that directly depends on the model density. In contrast, we *train* the model as a score-based model, but *use* it as a flow—both during training and inference—to generate samples. The model density is needed only to reweight the samples to approximate the target density. Since in principle the model used for resampling does not need to be the same as the model

---

**Algorithm 1:** Energy-based training epoch

**Input:** Boltzmann density $\tilde{p}$, training pairs $\{(G_i, L_i)\}_i$, torsional diffusion model $q$

**for each** $(G_i, L_i)$ **do**
    Sample $\boldsymbol{\tau}_1, \ldots \boldsymbol{\tau}_K \sim q_{G_i}(\boldsymbol{\tau} \mid L_i)$;
    **for** $k \leftarrow 1$ **to** $K$ **do**
        $\tilde{w}_k = \tilde{p}_{G_i}(\boldsymbol{\tau}_k \mid L_i)/q_{G_i}(\boldsymbol{\tau}_k \mid L_i)$;
    Approximate $J_{\text{DSM}}$ for $p_0 \propto \tilde{p}$ using $\{(\tilde{w}_i, \boldsymbol{\tau}_i)\}_i$;
    Minimize $J_{\text{DSM}}$;

---

being trained,[8] we can use very few steps (a shallow flow) during resampling to accelerate training, and then increase the number of steps (a deeper flow) for better approximations during inference—an option unavailable to existing Boltzmann generators.

## 4 Experiments

We evaluate torsional diffusion by comparing the generated and ground-truth conformers in terms of ensemble RMSD (Section 4.3) and properties (Section 4.4). Section 4.1 first discusses a preprocessing procedure required to train a conditional model $p_G(\boldsymbol{\tau} \mid L)$. Section 4.5 concludes with torsional Boltzmann generators. See Appendix H for additional results, including ablation experiments.

### 4.1 Conformer matching

In focusing on $p_G(\boldsymbol{\tau} \mid L)$, we have assumed that we can sample local structures $L \sim p_G(L)$ with RDKit. While this assumption is very good in terms of RMSD, the RDKit marginal $\hat{p}_G(L)$ is only an approximation of the ground truth $p_G(L)$. Thus, if we train on the denoising score-matching loss with ground truth conformers—i.e., conditioned on ground truth local structures—there will be a distributional shift at test time, where only approximate local structures from $\hat{p}_G(L)$ are available. We found that this shift significantly hurts performance.

We thus introduce a preprocessing procedure called *conformer matching*. In brief, for the *training* split only, we substitute each ground truth conformer $C$ with a synthetic conformer $\hat{C}$ with local structures $\hat{L} \sim \hat{p}_G(L)$ and made as similar as possible to $C$. That is, we use RDKit to generate

---

[8]For example, if the resampler were perfect, the procedure would reduce to normal denoising score matching.

Table 1: Quality of generated conformer ensembles for the GEOM-DRUGS test set in terms of Coverage (%) and Average Minimum RMSD (Å). We compute Coverage with a threshold of $\delta = 0.75$ Å to better distinguish top methods. Note that this is different from most prior works, which used $\delta = 1.25$ Å.

| | Recall | | | | Precision | | | |
|---|---|---|---|---|---|---|---|---|
| | Coverage ↑ | | AMR ↓ | | Coverage ↑ | | AMR ↓ | |
| Method | Mean | Med | Mean | Med | Mean | Med | Mean | Med |
| RDKit ETKDG | 38.4 | 28.6 | 1.058 | 1.002 | 40.9 | 30.8 | 0.995 | 0.895 |
| OMEGA | 53.4 | 54.6 | 0.841 | 0.762 | 40.5 | 33.3 | 0.946 | 0.854 |
| GeoMol | 44.6 | 41.4 | 0.875 | 0.834 | 43.0 | 36.4 | 0.928 | 0.841 |
| GeoDiff | 42.1 | 37.8 | 0.835 | 0.809 | 24.9 | 14.5 | 1.136 | 1.090 |
| Torsional Diffusion | **72.7** | **80.0** | **0.582** | **0.565** | **55.2** | **56.9** | **0.778** | **0.729** |

$\hat{L}$ and change torsion angles $\hat{\tau}$ to minimize $\text{RMSD}(C, \hat{C})$. Naively, we could sample $\hat{L} \sim \hat{p}_G(L)$ independently for each conformer, but this eliminates any possible dependence between $L$ and $\tau$ that could serve as training signal. Instead, we view the distributional shift as a domain adaptation problem that can be solved by optimally aligning $p_G(L)$ and $\hat{p}_G(L)$. See Appendix E for details.

## 4.2 Experimental setup

**Dataset** We evaluate on the GEOM dataset [Axelrod and Gómez-Bombarelli, 2022], which provides gold-standard conformer ensembles generated with metadynamics in CREST [Pracht et al., 2020]. We focus on GEOM-DRUGS—the largest and most pharmaceutically relevant part of the dataset— consisting of 304k drug-like molecules (average 44 atoms). To test the capacity to extrapolate to the largest molecules, we also collect from GEOM-MoleculeNet all species with more than 100 atoms into a dataset we call GEOM-XL and use it to evaluate models trained on DRUGS. Finally, we train and evaluate models on GEOM-QM9, a more established dataset but with significantly smaller molecules (average 11 atoms). Results for GEOM-XL and GEOM-QM9 are in Appendix H.

**Evaluation** We use the train/val/test splits from Ganea et al. [2021] and use the same metrics to compare the generated and ground truth conformer ensembles: Average Minimum RMSD (AMR) and Coverage. These metrics are reported both for Recall (R)—which measures how well the generated ensemble covers the ground-truth ensemble—and Precision (P)—which measures the accuracy of the generated conformers. See Appendix G for exact definitions and further details. Following the literature, we generate $2K$ conformers for a molecule with $K$ ground truth conformers.

**Baselines** We compare with the strongest existing methods from Section 2. Among cheminformatics methods, we evaluate RDKit ETKDG [Riniker and Landrum, 2015], the most established open-source package, and OMEGA [Hawkins et al., 2010, Hawkins and Nicholls, 2012], a commercial software in continuous development. Among machine learning methods, we evaluate GeoMol [Ganea et al., 2021] and GeoDiff [Xu et al., 2022], which have outperformed all previous models on the evaluation metrics. Note that GeoDiff originally used a small subset of the DRUGS dataset, so we retrained it using the splits from Ganea et al. [2021].

## 4.3 Ensemble RMSD

Torsional diffusion significantly outperforms all previous methods on GEOM-DRUGS (Table 1 and Figure 3), reducing by 30% the average minimum recall RMSD and by 16% the precision RMSD relative to the previous state-of-the-art method. Torsional diffusion is also the first ML method to consistently generate better ensembles than OMEGA. As OMEGA is a well-established product used in industry, this represents an essential step towards establishing the utility of conformer generation with machine learning.

Torsional diffusion offers specific advantages over both GeoDiff and GeoMol, the most advanced prior machine learning methods. GeoDiff, a Euclidean diffusion model, requires 5000 denoising steps to obtain the results shown, whereas our model—thanks to the reduced degrees of freedom—requires only 20 steps. In fact, our model outperforms GeoDiff with as few as 5 denoising steps. As seen in Table 2, this translates to enormous runtime improvements.

Figure 3: Mean coverage for recall (*left*) and precision (*right*) when varying the threshold value $\delta$ on GEOM-DRUGS.

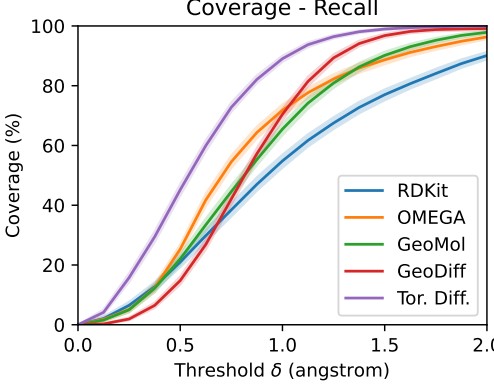
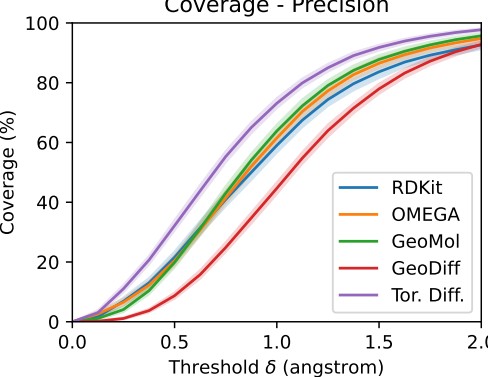

Table 2: Median AMR and runtime (core-secs per conformer) of machine learning methods, evaluated on CPU for comparison with RDKit.

| Method | Steps | AMR-R | AMR-P | Runtime |
|--------|-------|-------|-------|---------|
| RDKit | - | 1.002 | 0.895 | **0.10** |
| GeoMol | - | 0.834 | 0.841 | 0.18 |
| GeoDiff | 5000 | 0.809 | 1.090 | 305 |
| Torsional Diffusion | 5 | 0.685 | 0.963 | 1.76 |
| | 10 | 0.580 | 0.791 | 2.82 |
| | 20 | **0.565** | **0.729** | 4.90 |

Table 3: Median absolute error of generated v.s. ground truth ensemble properties. $E, \Delta\epsilon, E_{\min}$ in kcal/mol, $\mu$ in debye.

| Method | $E$ | $\mu$ | $\Delta\epsilon$ | $E_{\min}$ |
|--------|-----|-------|------------------|------------|
| RDKit | 0.81 | 0.52 | 0.75 | 1.16 |
| OMEGA | 0.68 | 0.66 | 0.68 | 0.69 |
| GeoMol | 0.42 | **0.34** | 0.59 | 0.40 |
| GeoDiff | 0.31 | 0.35 | 0.89 | 0.39 |
| Tor. Diff. | **0.22** | 0.35 | **0.54** | **0.13** |

Compared to torsional diffusion, GeoMol similarly makes use of intrinsic coordinates. However, since GeoMol can only access the molecular graph, it is less suited for reasoning about relationships that emerge only in a spatial embedding, especially between regions of the molecule that are distant on the graph. Our extrinsic-to-intrinsic score framework—which gives direct access to spatial relationships—addresses precisely this issue. The empirical advantages are most evident for the large molecules in GEOM-XL, on which GeoMol fails to improve consistently over RDKit (Appendix H). On the other hand, because GeoMol requires only a single-forward pass, it retains the advantage of faster runtime compared to diffusion-based methods.

### 4.4 Ensemble properties

While RMSD gives a *geometric* way to evaluate ensemble quality, we also consider the *chemical* similarity between generated and ground truth ensembles. For a random 100-molecule subset of DRUGS, we generate $\min(2K, 32)$ conformers per molecule, relax the conformers with GFN2-xTB [Bannwarth et al., 2019],[9] and compare the Boltzmann-weighted properties of the generated and ground truth ensembles. Specifically, the following properties are computed with xTB [Bannwarth et al., 2019]: energy $E$, dipole moment $\mu$, HOMO-LUMO gap $\Delta\epsilon$, and the minimum energy $E_{\min}$. The median errors for torsional diffusion and the baselines are shown in Table 4. Our method produces the most chemically accurate ensembles, especially in terms of energy. In particular, we significantly improve over GeoMol and GeoDiff in finding the lowest-energy conformers that are only (on median) 0.13 kcal/mol higher in energy than the global minimum.

---

[9]Results without relaxation (which are less chemically meaningful) are in Appendix H.

### 4.5 Torsional Boltzmann generator

Finally, we evaluate how well a torsional Boltzmann generator trained with MMFF [Halgren, 1996] energies can sample the corresponding Boltzmann density over torsion angles. We train and test on GEOM-DRUGS molecules with 3–7 rotatable bonds and use the local structures of the first ground-truth conformers. For the baselines, we implement annealed importance samplers (AIS) [Neal, 2001] with Metropolis-Hastings steps over the torsional space and tune the variance of the transition kernels.

Table 4 shows the quality of the samplers in terms of the *effective sample size* (ESS) given by the weights of 32 samples for each test molecule, which measures the $\alpha$-divergence (with $\alpha = 2$) between the model

Table 4: Effective sample size (out of 32) given by importance sampling weights over the torsional Boltzmann density.

| Method | Steps | Temp. (K) | | |
|---|---|---|---|---|
| | | 1000 | 500 | 300 |
| Uniform | – | 1.71 | 1.21 | 1.02 |
| AIS | 5 | 2.20 | 1.36 | 1.18 |
| | 20 | 3.12 | 1.76 | 1.30 |
| | 100 | 6.72 | 3.12 | 2.06 |
| Torsional BG | 5 | 7.28 | 3.60 | 3.04 |
| | 20 | **11.42** | **6.42** | **4.68** |

and Boltzmann distributions [Midgley et al., 2021]. Our method significantly outperforms the AIS baseline, and improves with increased step size despite being trained with only a 5-step resampler. Note that, since these evaluations are done on *unseen* molecules, they are beyond the capabilities of existing Boltzmann generators.

## 5 Conclusion

We presented *torsional diffusion*, a method for generating molecular conformers based on a diffusion process restricted to the most flexible degrees of freedom. Torsional diffusion is the first machine learning model to significantly outperform standard cheminformatics methods and is orders of magnitude faster than previous Euclidean diffusion models. Using the exact likelihoods provided by our model, we also train the first system-agnostic Boltzmann generator.

There are several exciting avenues for future work. A natural extension is to relax the rigid local structure assumption by developing an efficient diffusion-based model over the full space of intrinsic coordinates while still incorporating chemical constraints. Moreover, torsional diffusion—or similar ideas—could be applicable to larger molecular systems, for which fast, parsimonious models of structural flexibility could benefit applications such as drug discovery and protein design.

## Acknowledgments

We pay tribute to Octavian-Eugen Ganea (1987-2022), dear colleague, mentor, and friend without whom this work would have never been possible.

We thank Hannes Stärk, Wenxian Shi, Xiang Fu, Felix Faltings, Jason Yim, Adam Fisch, Alex Wu, Jeremy Wohlwend, Peter Mikhael, and Saro Passaro for helpful feedback and discussions. We thank Lagnajit Pattanaik, Minkai Xu, and Simon Axelrod for their advice and support when working with, respectively, GeoMol, GeoDiff and the GEOM dataset. This work was supported by the Machine Learning for Pharmaceutical Discovery and Synthesis (MLPDS) consortium, the Abdul Latif Jameel Clinic for Machine Learning in Health, the DTRA Discovery of Medical Countermeasures Against New and Emerging (DOMANE) threats program, the DARPA Accelerated Molecular Discovery program and the Sanofi Computational Antibody Design grant. We acknowledge support from the Department of Energy Computational Science Graduate Fellowship (BJ), the Robert Shillman Fellowship (GC), and the NSF Graduate Research Fellowship (JC).

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
