## A  Definitions

Consider a molecular graph $G = (\mathcal{V}, \mathcal{E})$ and its space of possible conformers $\mathcal{C}_G$. A conformer is an assignment $\mathcal{V} \mapsto \mathbb{R}^3$ of each atom to a point in 3D-space, defined up to global rototranslation. For notational convenience, we suppose there is an ordering of nodes such that we can regard a mapping as a vector in $\mathbb{R}^{3n}$ where $n = |\mathcal{V}|$. Then a conformer $C \in \mathcal{C}_G$ is a set of $SE(3)$-equivalent vectors in $\mathbb{R}^{3n}$—that is, $\mathcal{C}_G \cong \mathbb{R}^{3n}/SE(3)$. This defines the space of conformers in terms of *extrinsic* (or Cartesian) coordinates.

An *intrinsic* (or internal) coordinate is a function over $\mathcal{C}_G$—i.e., it is an $SE(3)$-invariant function over $\mathbb{R}^{3n}$. There are four types of such coordinates typically considered:

**Bond lengths**. For $(a, b) \in \mathcal{E}$, the bond length $l_{ab} \in [0, \infty)$ is defined as $|x_a - x_b|$.

**Bond angles**. For $a, b, c \in \mathcal{V}$ such that $a, c \in \mathcal{N}(b)$, the bond angle $\alpha_{abc} \in [0, \pi]$ is defined by

$$\cos \alpha_{abc} := \frac{(x_c - x_b) \cdot (x_a - x_b)}{|x_c - x_b||x_a - x_b|} \tag{11}$$

**Chirality**. For $a \in \mathcal{V}$ with 4 neighbors $b, c, d, e \in \mathcal{N}(a)$, the chirality $z_{abcd} \in \{-1, 1\}$ is defined as

$$z_{abcde} := \operatorname{sign} \det \begin{pmatrix} 1 & 1 & 1 & 1 \\ x_b - x_a & x_c - x_a & x_d - x_a & x_e - x_a \end{pmatrix} \tag{12}$$

Similar quantities are defined for atoms with other numbers of neighbors. Chirality is often considered part of the specification of the molecule, rather than the conformer. See Appendix F.3 for additional discussion on this point.

**Torsion angles**. For $(b, c) \in \mathcal{E}$, with a choice of reference neighbors $a \in \mathcal{N}(b) \setminus \{c\}, d \in \mathcal{N}(c) \setminus \{b\}$, the torsion angle $\tau_{abcd} \in [0, 2\pi)$ is defined as the dihedral angle between planes $abc$ and $bcd$:

$$\begin{aligned} \cos \tau_{abcd} &= \frac{\mathbf{n}_{abc} \cdot \mathbf{n}_{bcd}}{|\mathbf{n}_{abc}||\mathbf{n}_{bcd}|} \\ \sin \tau_{abcd} &= \frac{\mathbf{u}_{bc} \cdot (\mathbf{n}_{abc} \times \mathbf{n}_{bcd})}{|\mathbf{u}_{bc}||\mathbf{n}_{abc}||\mathbf{n}_{bcd}|} \end{aligned} \tag{13}$$

where $\mathbf{u}_{ab} = x_b - x_a$ and $\mathbf{n}_{abc}$ is the normal vector $\mathbf{u}_{ab} \times \mathbf{u}_{bc}$. Note that $\tau_{abcd} = \tau_{dcba}$—i.e., the dihedral angle is the same for four consecutively bonded atoms regardless of the direction in which they are considered.

A **complete set of intrinsic coordinates** of the molecule is a set of such functions $(f_1, f_2, \dots)$ such that $F(C) = (f_1(C), f_2(C), \dots)$ is a bijection. In other words, they fully specify a unique element of $\mathcal{C}_G$ without overparameterizing the space. In general there exist many possible such sets for a given molecular graph. We will not discuss further how to find such sets, as our work focuses on manipulating molecules in a way that holds fixed all $l, \alpha, z$ and only modifies (a subset of) torsion angles $\tau$.

As presently stated, the **torsion angle about a bond** $(b, c) \in \mathcal{E}$ is ill-defined, as it could be any $\tau_{abcd}$ with $a \in \mathcal{N}(b) \setminus \{c\}, d \in \mathcal{N}(c) \setminus \{b\}$. However, any complete set of intrinsic coordinates needs to only have at most one such $\tau_{abcd}$ for each bond $(b, c)$ [Ganea et al., 2021]. Thus, we often refer to *the* torsion angle about a bond $(b_i, c_i)$ as $\tau_i$ when reference neighbors $a_i, b_i$ are not explicitly stated.

## B  Propositions

### B.1  Torsion update

Given a freely rotatable bond $(b_i, c_i)$, by definition removing $(b_i, c_i)$ creates two connected components $\mathcal{V}(b_i), \mathcal{V}(c_i)$. Then, consider torsion angle $\tau_j$ at a different bond $(b_j, c_j)$ with neighbor choices $a_j \in \mathcal{N}(b_j), d_j \in \mathcal{N}(c_j), a_j \neq c_j, d_j \neq b_j$. Without loss of generality, there are two cases

- Case 1: $a_j, b_j, c_j, d_j \in \mathcal{V}(b_i)$
- Case 2: $d_j \in \mathcal{V}(c_i)$ and $a_j, b_j, c_j \in \mathcal{V}(b_i)$

Note that in Case 2, $c_j = b_i$ and $d_j = c_i$ must hold because there is only one edge between $\mathcal{V}(b_i), \mathcal{V}(c_i)$. With these preliminaries we now restate the proposition:

**Proposition 1.** *Let $(b_i, c_i)$ be a rotatable bond, let $\mathbf{x}_{\mathcal{V}(b_i)}$ be the positions of atoms on the $b_i$ side of the molecule, and let $R(\boldsymbol{\theta}, x_{c_i}) \in SE(3)$ be the rotation by Euler vector $\boldsymbol{\theta}$ about $x_{c_i}$. Then for $C, C' \in \mathcal{C}_G$, if $\tau_i$ is any definition of the torsion angle around bond $(b_i, c_i)$,*

$$
\begin{aligned}
\tau_i(C') &= \tau_i(C) + \theta \\
\tau_j(C') &= \tau_j(C) \quad \forall j \neq i
\end{aligned}
\qquad \text{if} \qquad \exists \mathbf{x} \in C, \mathbf{x}' \in C'. \quad
\begin{aligned}
\mathbf{x}'_{\mathcal{V}(b_i)} &= \mathbf{x}_{\mathcal{V}(b_i)} \\
\mathbf{x}'_{\mathcal{V}(c_i)} &= R(\theta\,\hat{\mathbf{r}}_{b_i c_i}, x_{c_i})\,\mathbf{x}_{\mathcal{V}(c_i)}
\end{aligned}
\tag{14}
$$

*where $\hat{\mathbf{r}}_{b_i c_i} = (x_{c_i} - x_{b_i})/\|x_{c_i} - x_{b_i}\|$.*

*Proof.* First we show $\tau_i(C') = \tau_i(C) + \theta$, for which it suffices to show $\tau_i(\mathbf{x}') = \tau_i(\mathbf{x}) + \theta$. Because $a_i, b_i \in \mathcal{V}(b_i)$, $x'_{a_i} = x_{a_i}$ and $x'_{b_i} = x_{b_i}$. Since the rotation of $\mathbf{x}_{\mathcal{V}(c_i)}$ is centered at $x_{c_i}$, we have $x'_{c_i} = x_{c_i}$ as well. Now we consider $d_i$ and $\mathbf{u}'_{cd} = x'_{d_i} - x_{c_i}$. By the Rodrigues rotation formula,

$$
\mathbf{u}'_{cd} = \mathbf{u}_{cd} \cos\theta + \frac{\mathbf{n}_{bcd}}{|\mathbf{u}_{bc}|} \sin\theta + \frac{\mathbf{u}_{bc}}{|\mathbf{u}_{bc}|}\left(\frac{\mathbf{u}_{bc}}{|\mathbf{u}_{bc}|} \cdot \mathbf{u}_{cd}\right)(1 - \cos\theta)
\tag{15}
$$

Then we have

$$
\mathbf{n}'_{bcd} = \mathbf{u}_{bc} \times \mathbf{u}'_{cd} = \mathbf{n}_{bcd} \cos\theta - \left(\mathbf{n}_{bcd} \times \frac{\mathbf{u}_{bc}}{|\mathbf{u}_{bc}|}\right)\sin\theta
\tag{16}
$$

To obtain $|\mathbf{n}'_{bcd}|$, note that since $\mathbf{n}_{bcd} \perp \mathbf{u}_{bc}$,

$$
\left|\mathbf{n}_{bcd} \times \frac{\mathbf{u}_{bc}}{|\mathbf{u}_{bc}|}\right| = |\mathbf{n}_{bcd}|
\tag{17}
$$

which gives $|\mathbf{n}'_{bcd}| = |\mathbf{n}_{bcd}|$. Thus,

$$
\begin{aligned}
\cos\tau'_i &= \frac{\mathbf{n}_{abc} \cdot \mathbf{n}'_{bcd}}{|\mathbf{n}_{abc}||\mathbf{n}_{bcd}|} = \frac{\mathbf{n}_{abc} \cdot \mathbf{n}_{bcd}}{|\mathbf{n}_{abc}||\mathbf{n}_{bcd}|}\cos\theta - \frac{\mathbf{n}_{abc} \cdot (\mathbf{n}_{bcd} \times \mathbf{u}_{bc})}{|\mathbf{n}_{abc}||\mathbf{n}_{bcd}||\mathbf{u}_{bc}|}\sin\theta \\
&= \cos\tau_i \cos\theta - \sin\tau_i \sin\theta = \cos(\tau_i + \theta)
\end{aligned}
\tag{18}
$$

Similarly,

$$
\begin{aligned}
\sin\tau'_i &= \frac{\mathbf{u}_{bc} \cdot (\mathbf{n}_{abc} \times \mathbf{n}'_{bcd})}{|\mathbf{u}_{bc}||\mathbf{n}_{abc}||\mathbf{n}_{bcd}|} = \frac{\mathbf{u}_{bc} \cdot (\mathbf{n}_{abc} \times \mathbf{n}_{bcd})}{|\mathbf{u}_{bc}||\mathbf{n}_{abc}||\mathbf{n}_{bcd}|}\cos\theta - \frac{\mathbf{u}_{bc} \cdot (\mathbf{n}_{abc} \times (\mathbf{n}_{bcd} \times \mathbf{u}_{bc}))}{|\mathbf{u}_{bc}|^2|\mathbf{n}_{abc}||\mathbf{n}_{bcd}|}\sin\theta \\
&= \sin\tau_i \cos\theta + \cos\tau_i \sin\theta = \sin(\tau_i + \theta)
\end{aligned}
\tag{19}
$$

Therefore, $\tau'_i = \tau_i + \theta$

Now we show $\tau'_j = \tau_j$ for all $j \neq i$. Consider any such $j$. For Case 1, $x'_{a_j} = x_{a_j}, x'_{b_j} = x_{b_j}, x'_{c_j} = x_{c_j}, x'_{d_j} = x_{d_j}$ so clearly $\tau'_j = \tau_j$. For Case 2, $x'_{a_j} = x_{a_j}, x'_{b_j} = x_{b_j}, x'_{c_j} = x_{c_j}$ immediately. But because $d_j = c_i$, we also have $x'_{d_j} = x_{d_j}$. Thus, $\tau'_j = \tau_j$. $\square$

## B.2 Parity equivariance

**Proposition 2.** *If $p(\boldsymbol{\tau}(C) \mid L(C)) = p(\boldsymbol{\tau}(-C) \mid L(-C))$, then for all diffusion times $t$,*

$$
\nabla_{\boldsymbol{\tau}} \log p_t(\boldsymbol{\tau}(C) \mid L(C)) = -\nabla_{\boldsymbol{\tau}} \log p_t(\boldsymbol{\tau}(-C) \mid L(-C))
\tag{20}
$$

*Proof.* From Equation 13 we see that for any torsion $\tau_i$, we have $\tau_i(-C) = -\tau_i(C)$; therefore $\tau_i(-C) = -\tau_i(C)$, which we denote $\boldsymbol{\tau}_-$. Also denote $\boldsymbol{\tau} := \boldsymbol{\tau}(C), p_t(\boldsymbol{\tau}) := p_t(\boldsymbol{\tau} \mid L(C))$ and $p'_t(\boldsymbol{\tau}_-) := p_t(\boldsymbol{\tau}_- \mid L(-C))$. We claim $p_t(\boldsymbol{\tau}) = p'_t(\boldsymbol{\tau}_-)$ for all $t$. Since the perturbation kernel (equation 3) is parity invariant,

$$
\begin{aligned}
p'_t(\boldsymbol{\tau}_-) &= \int_{\mathbb{T}^m} p'_0(\boldsymbol{\tau}'_-)p_{t|0}(\boldsymbol{\tau}_- \mid \boldsymbol{\tau}'_-)\, d\boldsymbol{\tau}'_- \\
&= \int_{\mathbb{T}^m} p_0(\boldsymbol{\tau}')p_{t|0}(\boldsymbol{\tau} \mid \boldsymbol{\tau}')\, d\boldsymbol{\tau}'_- = p_t(\boldsymbol{\tau})
\end{aligned}
\tag{21}
$$

Next, we have

$$
\begin{aligned}
\nabla_{\boldsymbol{\tau}} \log p'_t(\boldsymbol{\tau}_-) &= \frac{\partial \boldsymbol{\tau}_-}{\partial \boldsymbol{\tau}} \nabla_{\boldsymbol{\tau}_-} \log p'_t(\boldsymbol{\tau}_-) \\
&= -\nabla_{\boldsymbol{\tau}} \log p_t(\boldsymbol{\tau})
\end{aligned}
\tag{22}
$$

which concludes the proof. $\square$

## B.3 Likelihood conversion

**Proposition 3.** *Let $\mathbf{x} \in C(\boldsymbol{\tau}, L)$ be a centered conformer in Euclidean space. Then,*

$$p_G(\mathbf{x} \mid L) = \frac{p_G(\boldsymbol{\tau} \mid L)}{8\pi^2 \sqrt{\det g}} \quad \text{where} \ \ g_{\alpha\beta} = \sum_{k=1}^{n} J_\alpha^{(k)} \cdot J_\beta^{(k)} \tag{23}$$

*where the indices $\alpha, \beta$ are integers between 1 and $m+3$. For $1 \leq \alpha \leq m$, $J_\alpha^{(k)}$ is defined as*

$$J_i^{(k)} = \tilde{J}_i^{(k)} - \frac{1}{n} \sum_{\ell=1}^{n} \tilde{J}_i^{(\ell)} \quad \text{with} \ \ \tilde{J}_i^{(\ell)} = \begin{cases} 0 & \ell \in \mathcal{V}(b_i), \\ \frac{\mathbf{x}_{b_i} - \mathbf{x}_{c_i}}{||\mathbf{x}_{b_i} - \mathbf{x}_{c_i}||} \times (\mathbf{x}_\ell - \mathbf{x}_{c_i}), & \ell \in \mathcal{V}(c_i), \end{cases} \tag{24}$$

*and for $\alpha \in \{m+1, m+2, m+3\}$ as*

$$J_{m+1}^{(k)} = \mathbf{x}_k \times \hat{x}, \qquad J_{m+2}^{(k)} = \mathbf{x}_k \times \hat{y}, \qquad J_{m+3}^{(k)} = \mathbf{x}_k \times \hat{z}, \tag{25}$$

*where $(b_i, c_i)$ is the freely rotatable bond for torsion angle $i$, $\mathcal{V}(b_i)$ is the set of all nodes on the same side of the bond as $b_i$, and $\hat{x}, \hat{y}, \hat{z}$ are the unit vectors in the respective directions.*

*Proof.* Let $M$ be $(m+3)$-dimensional manifold embedded in $3n$-dimensional Euclidean space formed by the set of all centered conformers with fixed local structures but arbitrary torsion angles and orientation. A natural set of coordinates for $M$ is $q^\alpha = \{\tau_1, \tau_2, \ldots, \tau_m, \omega_x, \omega_y, \omega_z\}$, where $\tau_i$ is the torsion angle at bond $i$ and $\omega_x, \omega_y, \omega_z$ define the global rotation about the center of mass:

$$\mathbf{x}_k = \tilde{\mathbf{x}}_k - \frac{1}{n} \sum_{\ell=1}^{n} \tilde{\mathbf{x}}_\ell \quad \text{where} \ \ \tilde{\mathbf{x}}_\ell = e^{\Lambda(\omega)} \mathbf{x}_k', \quad \Lambda(\omega) = \begin{pmatrix} 0 & -\omega_z & \omega_y \\ \omega_z & 0 & -\omega_x \\ -\omega_y & \omega_x & 0 \end{pmatrix}. \tag{26}$$

Here $\mathbf{x}_k'$ is the position of atom $k$ as determined by the torsion angles, without centering or global rotations, and $\omega_x, \omega_y, \omega_z$ are rotation about the $x$, $y$, and $z$ axis respectively.

Consider the set of covariant basis vectors

$$\mathbf{J}_\alpha = \frac{\partial \mathbf{x}}{\partial q^\alpha}. \tag{27}$$

and corresponding the covariant components of the metric tensor,

$$g_{\alpha\beta} = \mathbf{J}_\alpha \cdot \mathbf{J}_\beta = \frac{\partial \mathbf{x}}{\partial q^\alpha} \cdot \frac{\partial \mathbf{x}}{\partial q^\beta}. \tag{28}$$

The conversion factor between torsional likelihood and Euclidean likelihood is given by

$$\int \sqrt{\det \mathbf{g}} \, d^3\omega, \tag{29}$$

where $\sqrt{\det \mathbf{g}} \, d^{m+3}q$ is the invariant volume element on $M$ [Carroll, 2019], and the integration over $\omega$ marginalizes over the uniform distribution over global rotations. The calculation of Eq. 29 proceeds as follows.

Let the position of the $k$'th atom be $x_k$, and let the three corresponding components of $\mathbf{J}_\alpha$ be $J_\alpha^{(k)}$. For $1 \leq i \leq m$, $J_i^{(k)}$ is given by

$$J_i^{(k)} = \frac{\partial}{\partial \tau_i} \left( \tilde{\mathbf{x}}_k - \frac{1}{n} \sum_{\ell=1}^{n} \tilde{\mathbf{x}}_\ell \right) = \tilde{J}_i^{(k)} - \frac{1}{n} \sum_{\ell=1}^{n} \tilde{J}_i^{(\ell)} \tag{30}$$

where $\tilde{J}_i^{(k)} := \partial \tilde{\mathbf{x}}_k / \partial \tau_i$ is the displacement of atom $k$ upon an infinitesmal change in the torsion angle $\tau_i$, without considering the change in the center of mass. Clearly $\tilde{J}_i^{(b_i)} = \tilde{J}_i^{(c_i)} = 0$ because neither $b_i$ nor $c_i$ itself is displaced; furthermore, all atoms on the $b$ side of torsioning bond are not displaced, so $J_i^{(k)} = 0$ for all $k \in \mathcal{N}(b_i)$. The remaining atoms, in $\mathcal{N}(c_i)$, are rotated about the axis of the $(b_i, c_i)$ bond. The displacement per infinitesimal $\partial \tau_i$ is given by the cross product of the unit normal along the rotation axis, $(\tilde{x}_{c_i} - \tilde{x}_{b_i})/||\tilde{x}_{c_i} - \tilde{x}_{b_i}||$, with the displacement from rotation axis, $\tilde{x}_k - \tilde{x}_{b_i}$. This cross product yields $J_\alpha^{(k)}$ in Eq. 24, where the tildes are dropped as relative positions do not depend on center of mass. For $\alpha \in \{m+1, m+2, m+3\}$, a similar consideration of the cross product with the rotation axis yields Eq. 25. Finally, since none of the components of the metric tensor depend explicitly on $\omega$, the integration over $\omega$ in Eq. 29 is trivial and yields the volume over $SO(3)$ of $8\pi^2$ [Chirikjian, 2011], proving the proposition. $\square$

## C  Training and inference procedures

Algorithms 2 and 3 summarize, respectively, the training and inference procedures used for torsional diffusion. In practice, during training, we limit $K_G$ to 30 i.e. we only consider the first 30 conformers found by CREST (typically those with the largest Boltzmann weight). Moreover, molecules are batched and an Adam optimizer with a learning rate scheduler is used for optimization. For inference, to fairly compare with other methods from the literature, we follow Ganea et al. [2021] and set $K$ to be twice the number of conformers returned by CREST.

---

**Algorithm 2:** Training procedure

---

**Input:** molecules $[G_0, ..., G_N]$ each with true conformers $[C_{G,1}, ...C_{G,K_G}]$, learning rate $\alpha$
**Output:** trained score model $\mathbf{s}_\theta$
conformer matching process for each $G$ to get $[\hat{C}_{G,1}, ...\hat{C}_{G,K_G}]$;
**for** $epoch \leftarrow 1$ **to** $epoch_{\max}$ **do**
    **for** $G$ **in** $[G_0, ..., G_N]$ **do**
        sample $t \in [0, 1]$ and $\hat{C} \in [\hat{C}_{G,1}, ...\hat{C}_{G,K_G}]$;
        sample $\Delta\boldsymbol{\tau}$ from wrapped normal $p_{t|0}(\cdot \mid \mathbf{0})$ with $\sigma = \sigma_{\min}^{1-t} \sigma_{\max}^t$;
        apply $\Delta\boldsymbol{\tau}$ to $\hat{C}$;
        predict $\delta\boldsymbol{\tau} = \mathbf{s}_{\theta,G}(\hat{C}, t)$;
        update $\theta \leftarrow \theta - \alpha\nabla_\theta \|\delta\boldsymbol{\tau} - \nabla_{\Delta\boldsymbol{\tau}} p_{t|0}(\Delta\boldsymbol{\tau} \mid \mathbf{0})\|^2$;

---

**Algorithm 3:** Inference procedure

---

**Input:** molecular graph $G$, number conformers $K$, number steps $N$
**Output:** predicted conformers $[C_1, ...C_K]$
generate local structures by obtaining conformers $[C_1, ...C_K]$ from RDKit;
**for** $C$ **in** $[C_1, ...C_K]$ **do**
    sample $\Delta\boldsymbol{\tau} \sim U[0, 2\pi]^m$ and apply to $C$ to randomize torsion angles;
    **for** $n \leftarrow N$ **to** 1 **do**
        let $t = n/N$, $g(t) = \sigma_{\min}^{1-t} \sigma_{\max}^t \sqrt{2\ln(\sigma_{\max}/\sigma_{\min})}$;
        predict $\delta\boldsymbol{\tau} = \mathbf{s}_{\theta,G}(\hat{C}, t)$;
        draw $\mathbf{z}$ from wrapped normal with $\sigma^2 = 1/N$;
        set $\Delta\boldsymbol{\tau} = (g^2(t)/N) \delta\boldsymbol{\tau} + g(t) \mathbf{z}$;
        apply $\Delta\boldsymbol{\tau}$ to $C$;

---

## D  Score network architecture

**Overview**  To perform the torsion score prediction under these symmetry constraints we design an architecture formed by three components: an embedding layer, a series of $K$ interaction layers and a pseudotorque layer. The pseudotorque layer produces pseudoscalar torsion scores $\delta\tau := \partial \log p/\partial\tau$ for every rotatable bond. Following the notation from Thomas et al. [2018] and Batzner et al. [2022], we represent the node representations as $V_{acm}^{(k,l,p)}$ a dictionary with keys the layer $k$, rotation order $l$ and parity $p$ that contains tensors with shapes $[|\mathcal{V}|, n_l, 2l+1]$ corresponding to the indices of the node, channel and representation respectively. We use the `e3nn` library [Geiger et al., 2022] to implement our architecture.

**Embedding layer**  In the embedding layer, we build a radius graph $(\mathcal{V}, \mathcal{E}_{r_{\max}})$ around each atom on top of the original molecular graph and generate initial scalar embeddings for nodes $V_a^{(0,0,1)}$ and edges $e_{ab}$ combining chemical properties, sinusoidal embeddings of time $\phi(t)$ [Vaswani et al., 2017]

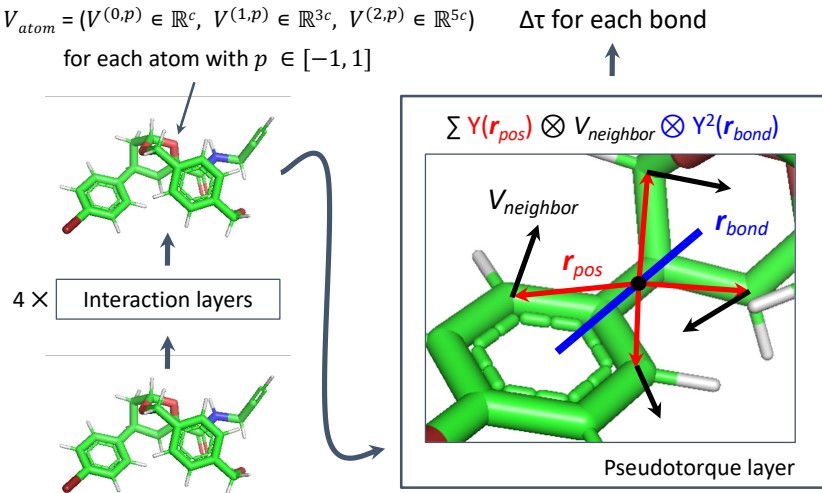

$V_{atom} = (V^{(0,p)} \in \mathbb{R}^c,\ V^{(1,p)} \in \mathbb{R}^{3c},\ V^{(2,p)} \in \mathbb{R}^{5c})$     $\Delta\tau$ for each bond

for each atom with $p \in [-1, 1]$

$\sum \Upsilon(\mathbf{r}_{pos}) \otimes V_{neighbor} \otimes \Upsilon^2(\mathbf{r}_{bond})$

$V_{neighbor}$

$\mathbf{r}_{bond}$

$\mathbf{r}_{pos}$

$4 \times$   Interaction layers

Pseudotorque layer

Figure 4: Overview of the architecture and visual intuition of the pseudotorque layer.

and, for the edges, a radial basis function representation of their length $\mu(r_{ab})$ [Schütt et al., 2017]:

$$\mathcal{E}_{r_{\max}} = \mathcal{E} \sqcup \{(a,b) \mid r_{ab} < r_{\max}\}$$
$$e_{ab} = \Upsilon^{(e)}(fe_{ab}||\mu(r_{ab})||\phi(t)) \quad \forall (a,b) \in \mathcal{E}_{r_{\max}} \tag{31}$$
$$V_a^{(0,0,1)} = \Upsilon^{(v)}(f_a||\phi(t)) \quad \forall a \in \mathcal{V}$$

where $\Upsilon^{(e)}$ and $\Upsilon^{(v)}$ are learnable two-layers MLPs, $r_{ab}$ is the Euclidean distance between atoms $a$ and $b$, $r_{\max} = 5$ Å is the distance cutoff, $f_a$ are the chemical features of atom $a$, $f_{ab}$ are the chemical features of bond $(a,b)$ if it was part of $\mathcal{E}$ and 0 otherwise.

The node and edge chemical features $f_a$ and $f_{ab}$ are constructed as in Ganea et al. [2021]. Briefly, the node features include atom identity, atomic number, aromaticity, degree, hybridization, implicit valence, formal charge, ring membership, and ring size, constituting a 74-dimensional vector for GEOM-DRUGS and 44-dimensional for QM9 (due to fewer atom types). The edge features are a 4 dimensional one-hot encoding of the bond type.

**Interaction layers** The interaction layers are based on E(3)NN [Geiger et al., 2022] convolutional layers. At each layer, for every pair of nodes in the graph, we construct messages using tensor products of the current irreducible representation of each node with the spherical harmonic representations of the normalized edge vector. These messages are themselves irreducible representations, which are weighted channel-wise by a scalar function of the current scalar representations of the two nodes and the edge and aggregated with Clebsch-Gordan coefficients.

At every layer $k$, for every node $a$, rotation order $l_o$, and output channel $c'$:

$$V_{ac'm_o}^{(k,l_o,p_o)} = \sum_{l_f, l_i, p_i} \sum_{m_f, m_i} C_{(l_i,m_i)(l_f,m_f)}^{(l_o,m_o)} \frac{1}{|\mathcal{N}_a|} \sum_{b \in \mathcal{N}_a} \sum_c \psi_{abc}^{(k,l_o,l_f,l_i,p_i)} Y_{m_f}^{(l_f)}(\hat{r}_{ab}) V_{bcm_i}^{(k-1,l_i,p_i)}$$

$$\text{with } \psi_{abc}^{(k,l_o,l_f,l_i,p_i)} = \Psi_c^{(k,l_o,l_f,l_i,p_i)}(e_{ab}||V_a^{(k-1,0,1)}||V_b^{(k-1,0,1)}) \tag{32}$$

where the outer sum is over values of $l_f, l_i, p_i$ such that $|l_i - l_f| \le l_o \le l_i + l_f$ and $(-1)^{l_f} p_i = p_o$, $C$ indicates the Clebsch-Gordan coefficients [Thomas et al., 2018], $\mathcal{N}_a = \{b \mid (a,b) \in \mathcal{E}_{\max}\}$ the neighborhood of $a$ and $Y$ the spherical harmonics. The rotational order of the nodes representations $l_o$ and $l_i$ and of the spherical harmonics of the edges ($l_f$) are restricted to be at most 2. All the learnable weights are contained in $\Psi$, a dictionary of MLPs that compute per-channel weights based on the edge embeddings and scalar features of the outgoing and incoming node.

**Pseudotorque layer** The final part of our architecture is a pseudotorque layer that predicts a pseudoscalar score $\delta\tau$ for each rotatable bond from the per-node outputs of the interaction layers.

For every rotatable bond, we construct a tensor-valued filter, centered on the bond, from the tensor product of the spherical harmonics with a $l = 2$ representation of the *bond axis*. Since the parity of the $l = 2$ spherical harmonic is even, this representation does not require a choice of bond direction. The filter is then used to convolve with the representations of every neighbor on a radius graph, and the products which produce pseudoscalars are passed through odd-function (i.e., with tanh nonlinearity and no bias) dense layers (not shown in equation 33) to produce a single prediction.

For all rotatable bonds $g = (g_0, g_1) \in \mathcal{E}_{\text{rot}}$ and $b \in \mathcal{V}$, let $r_{gb}$ and $\hat{r}_{gb}$ be the magnitude and direction of the vector connecting the center of bond $g$ and $b$.

$$
\begin{aligned}
\mathcal{E}_\tau &= \{(g, b) \mid g \in \mathcal{E}_r, b \in \mathcal{V}, r_{gb} < r_{\max}\} \qquad e_{gb} = \Upsilon^{(\tau)}(\mu(r_{gb})) \\
T_{gbm_o}^{(l_o, p_o)} &= \sum_{m_g, m_r, l_r : p_o = (-1)^{l_r}} C_{(2, m_g)(l_r, m_r)}^{(l_o, m_o)} Y_{m_f}^{(2)}(\hat{r}_g) \, Y_{m_r}^{(l_r)}(\hat{r}_{gb}) \\
\delta\tau_g &= \sum_{l, p_f, p_i : p_f p_i = -1} \sum_{m_o, m_i} C_{(l, m_f)(l, m_i)}^{(0,0)} \frac{1}{|\mathcal{N}_g|} \sum_{b \in \mathcal{N}_g} \sum_c \gamma_{gcb}^{(l, p_i)} \, T_{gbm_f}^{(l, p_f)} \, V_{bcm_i}^{(K, l, p_i)} \\
&\qquad \text{with } \gamma_{gcb}^{(l, p_i)} = \Gamma_c^{(l, p_i)}(e_{gb} || V_b^{(K,0,1)} || V_{g_0}^{(K,0,1)} + V_{g_1}^{(K,0,1)})
\end{aligned}
\tag{33}
$$

where $\Upsilon^{(\tau)}$ and $\Gamma$ are MLPs with learnable parameters and $\mathcal{N}_g = \{b \mid (g, b) \in \mathcal{E}_\tau\}$.

# E   Conformer matching

The conformer matching procedure, summarised in Algorithm 4, proceeds as follows. For a molecule with $K$ conformers, we first generate $K$ random local structure estimates $\hat{L}$ from RDKit. To match with the ground truth local structures, we compute the cost of matching each true conformer $C$ with each estimate $\hat{L}$ (i.e. a $K \times K$ cost matrix), where the cost is the best RMSD that can be achieved by modifying the torsions of the RDKit conformer with local structure $\hat{L}$ to match the ground truth conformer $C$. Note that in practice, we compute an upper bound to this optimal RMSD using the fast von Mises torsion matching procedure proposed by Stärk et al. [2022].

We then find an optimal matching of true conformers $C$ to local structure estimates $\hat{L}$ by solving the linear sum assignment problem over the approximate cost matrix [Crouse, 2016]. Finally, for each matched pair, we find the true optimal $\hat{C}$ by running a differential evolution optimization procedure over the torsion angles [Méndez-Lucio et al., 2021]. The complete assignment resulting from the linear sum solution guarantees that there is no distributional shift in the local structures seen during training and inference.

---

**Algorithm 4:** Conformer matching

**Input:** true conformers of $G$ $[C_1, ... C_K]$
**Output:** approximate conformers for training $[\hat{C}_1, ... \hat{C}_K]$
generate local structures $[\hat{L}_1, ... \hat{L}_K]$ with RDKit;
**for** $(i, j)$ **in** $[1, K] \times [1, K]$ **do**
$\quad$ $C_{\text{temp}} = \text{von\_Mises\_matching}(C_i, \hat{L}_j)$;
$\quad$ cost[i,j] = $\text{RMSD}(C_i, C_{\text{temp}})$;
assignment = linear_sum_assignment(cost);
**for** $i \leftarrow 1$ **to** $K$ **do**
$\quad$ j = assignment[i];
$\quad$ $\hat{C}_i = \text{differential\_evolution}(C_i, \hat{L}_j, \text{RMSD})$;

---

Table 5 shows the average RMSD between a ground truth conformer $C_i$ and its matched conformer $\hat{C}_i$. The average RMSD of 0.324 Å obtained via conformer matching provides an approximate lower bound on the achievable AMR performance for methods that do not change the local structure and take those from RDKit (further discussion in Appendix F.1).

Table 5: Average $\mathrm{RMSD}(c_i, \hat{c}_i)$ achieved by different variants of conformer matching. "Original RDKit" refers to the RMSD between a random RDKit conformer and a ground truth conformer without any optimization. In "Von Mises optimization" and "Differential evolution," the torsions of the RDKit conformer are adjusted using the respective procedures, but the pairing of RDKit and ground truth conformers is still random. In "Conformer matching," the cost-minimizing assignment prior to differential evolution provides a 15% improvement in average RMSD. The results are shown for a random 300-molecule subset of GEOM-DRUGS.

| Matching method | RMSD (Å) |
| --- | --- |
| Original RDKit | 1.448 |
| Von Mises optimization | 0.728 |
| Differential evolution | 0.379 |
| Conformer matching | 0.324 |

# F   Additional discussion

## F.1   RDKit local structures

In this section, we provide empirical justification for the claim that cheminformatics methods like RDKit already provide accurate local structures. It is well known in chemistry that bond lengths and angles take on a very narrow range of values due to strong energetic constraints. However, it is not trivial to empirically evaluate the claim due to the difficulty in defining a distance measure between a pair of local structures. In this section, we will employ two sets of observations: marginal error distributions and matched conformer RMSD.

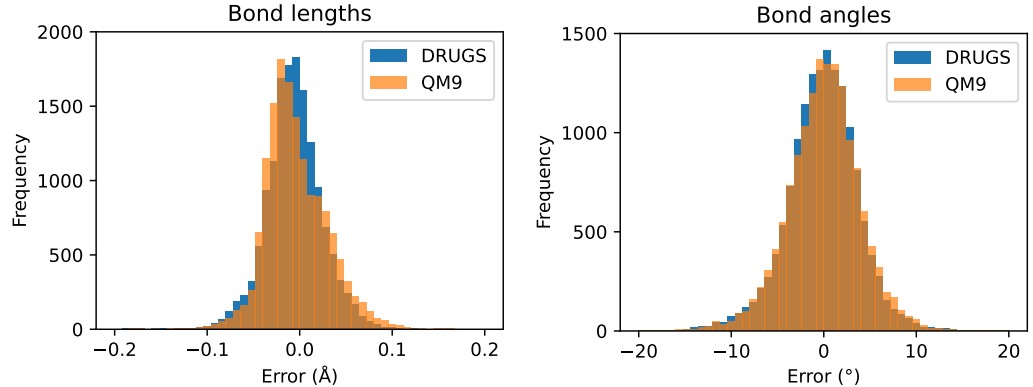

Figure 5: Histogram of the errors in 15000 predicted bond lengths and angles from randomly sampled molecules in GEOM-DRUGS and GEOM-QM9.

**Marginal error distributions**   We examine the distribution of errors of the bond lengths and angles in a random RDKit conformer relative to the corresponding lengths and angles in a random CREST conformer (Figure 5). The distributions are narrow and uni-modal distributions around zero, with a RMSE of 0.03 Å for bond lengths and 4.1° for bond angles on GEOM-DRUGS. Comparing DRUGS and QM9, the error distribution does not depend on the size of the molecule. Although it is difficult to determine how these variations will compound or compensate for each other in the global conformer structure, the analysis demonstrates that bond lengths and angles have little flexibility (i.e., no strong variability among conformers) and are accurately predicted by RDKit.

**Matched conformer RMSD**   We can more rigorously analyze the quality of a local structure $\hat{L}$ with respect to a given reference conformer $C$ by computing the minimum RMSD that can be obtained by combining $\hat{L}$ with optimal torsion angles. That is, we consider the RMSD distance of $C$ to the closest point on the manifold of possible conformers with local structure $\hat{L}$: $\mathrm{RMSD}_{\min}(C, \hat{L}) := \min_\tau \mathrm{RMSD}(C, \hat{C})$ where $\hat{C} = (\hat{L}, \tau)$.

Conveniently, $\hat{C}$ is precisely the output of the differential evolution in Appendix E. Thus, the average RSMD reported in the last row of Table 5 is the expected $\mathrm{RMSD}_{\min}$ of an optimal assignment of RDKit local structures to ground-truth conformers. This distance—0.324 Å on GEOM-DRUGS—is significantly smaller than the error of the current state-of-the-art conformer generation methods. Further, it is only slightly larger than the average $\mathrm{RMSD}_{\min}$ of 0.284 Å resulting from matching a ground truth conformer to the local structure of another randomly chosen ground truth conformer, which provides a measure of the variability among ground truth local structures. These observations support the claim that the accuracy of existing approaches on drug-like molecules can be significantly improved via better conditional sampling of torsion angles.

## F.2  Torsion updates

In the main text, we viewed updates $\Delta\tau$ as changes to a torsion angle $\tau$, and asserted that the same update applied to any torsion angle at a given bond (i.e., with any choice of reference neighbors) results in the same conformer. Given this, a potentially more intuitive presentation is to *define* $\Delta\tau$ for a bond as a relative rotation around that bond, without reference to any torsion angle.

Consider a rotatable bond $(b, c)$ and the connected components $\mathcal{V}(b), \mathcal{V}(c)$ formed by removing the bond. Let $\hat{\mathbf{r}}_{bc} = (x_c - x_b)/|x_c - x_b|$ and similarly $\hat{\mathbf{r}}_{cb} = -\hat{\mathbf{r}}_{bc}$. Because the bond is freely rotatable, consider rotations of each side of the molecule around the bond axis given by $\hat{\mathbf{r}}_{bc}$. Specifically, let $\mathcal{V}(b)$ be rotated by some Euler vector $\boldsymbol{\theta}_b := \theta_b \hat{\mathbf{r}}_{bc}$ around $x_b$, and $\mathcal{V}(c)$ by $\boldsymbol{\theta}_c := \theta_c \hat{\mathbf{r}}_{bc}$ around $x_c$. Then the rotations induce a *torsion update* $\Delta\tau$ if $\theta_c - \theta_b = \Delta\tau$; or equivalently

$$\Delta\tau = (\boldsymbol{\theta}_c - \boldsymbol{\theta}_b) \cdot \hat{\mathbf{r}}_{bc} \tag{34}$$

The expression remains unchanged if we swap the indices $b, c$; thus there is no sign ambiguity. Some less formal but possibly more intuitive restatements of the sign convention are:

- Looking down a bond, a *positive* update is given by a CCW rotation of the nearer side; or a CW rotation of the further side

- For a viewer positioned in the middle of the bond, a *positive* update is given by the CW rotation of any one side

- A *positive* update is given by Euler vectors that point *outwards* from the bond

These are illustrated in Figure 6.

Since the Euler vector $\boldsymbol{\theta}$ is a pseudovector that remains unchanged under parity inversion, while $\hat{\mathbf{r}}$ is a normal vector, it is apparent that $\Delta\tau$—and any model predicting $\Delta\tau$—must be a pseudoscalar.

Because the update is determined by a relative rotation, it is not necessary to specify which side to rotate. That is, the same torsion update can be accomplished by rotating only one side, both sides in opposite directions, or both sides in the same direction. In practical implementation, we rotate the side of the molecule with fewer atoms, and keep the other side fixed.

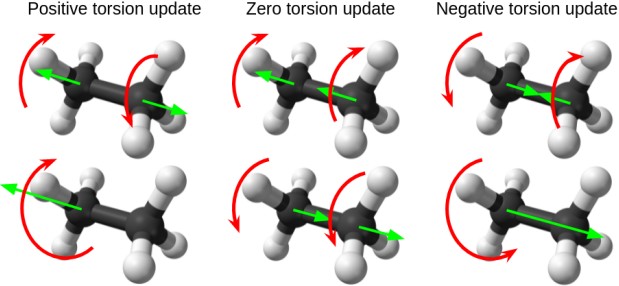

Figure 6: Torsion updates as relative rotations, with the rotations shown with curved red arrows and Euler vectors shown with straight green arrows. The second row emphasizes that the sign convention and update depend only on the *relative* motion of the two sides.

### F.3 Chemical isomerism

We have defined a molecule in terms of its bond connectivity, i.e., as a graph $G = (\mathcal{V}, \mathcal{E})$ with atoms $\mathcal{V}$ and $\mathcal{E}$. In chemistry, however, it is universal to consider molecules with the same connectivity, but whose conformers cannot interconvert, as different molecules called *stereoisomers*. In our formalism, stereoisomers correspond to subsets of the space of conformers $\mathcal{C}_G$ for some molecular graph $G$. Many types of stereoisomerism exist, but the two most important are:

- **Chirality**. Conformers with distinct values of chirality tags $\{z_i\}$—one for each chiral atom meeting certain criteria—are considered different molecules.
- **E/Z isomerism**, also called cis/trans isomerism. For each double bond meeting certain criteria, the space $[0, 2\pi)$ is partitioned into two halves, such that conformers are considered different molecules depending on the value of the torsion angle.

These are not meant to be formal definitions, and we refer to standard chemistry texts for a more detailed treatment. For our purposes, the key implication is that conformer generation requires generating conformers consistent with a *given stereoisomer*. For a molecular graph $G$ with $k$ relevant chiral centers and $l$ relevant double bonds, there are $2^{k+l}$ possible stereoisomers, corresponding to the partition of $\mathcal{C}_G$ into $2^{k+l}$ disjoint subsets—only one of which corresponds to the molecule under consideration.

Torsional diffusion automatically handles chirality. Because we have considered chirality to be part of the local structure, it is drawn from the cheminformatics package RDKit, which is given the full identification of the stereoisomer along with the molecular graph, and is not modified by the torsional diffusion. Hence, our method always generates conformers with the correct chirality at each chiral center. On the other hand, GeoDiff does not consider chirality at all, while GeoMol generates molecules without any chirality constraints, and merely inverts the chiral centers that were generated incorrectly.

E/Z isomerism is significantly trickier, as it places a constraint on the torsion angles at double bonds, which are considered freely rotatable in our framework. Presently, torsional diffusion does not attempt to capture E/Z isomerism. One possible way of doing so is to augment the molecular graph with edges of a special type, and we leave such augmentation to future work. GeoDiff and GeoMol also do not attempt to treat E/Z isomerism.

More generally, while the abstract view of molecules as graphs has enabled rapid advances in molecular machine learning, stereoisomerism shows that it is clearly a simplification. As stereoisomers can have significantly different chemical properties and bioactivities, a more complete view of molecular space will be essential for further advances in molecular machine learning.

### F.4 Limitations of torsional diffusion

As demonstrated in Section 4, torsional diffusion significantly improves the accuracy and reduces the denoising runtime for conformer generation. However, torsional diffusion also has a number of limitations that we will discuss in this section.

**Conformer generation**   The first clear limitation is that the error that torsional diffusion can achieve is lower bounded by the *quality of the local structure* from the selected cheminformatics method. As discussed in Appendix F.1, this corresponds to the mean RMSD obtained after conformer matching, which is 0.324 Å with RDKit local structures on DRUGS. Moreover, due to the the local structure *distributional shift* discussed in Section 4.1, conformer matching (or another method bridging the shift) is required to generate the training set. However, the resulting conformers are not the minima of the (unconditional or even conditional) potential energy function. Thus, the learning task becomes less physically interpretable and potentially more difficult; empirically we observe this clearly in the training and validation score-matching losses. We leave to future work the exploration of *relaxations* of the rigid local structures assumption in a way that would still leverage the predominance of torsional flexibility in molecular structures, while at the same time allowing some flexibility in the independent components.

**Rings** The largest source of flexibility in molecular conformations that is not directly accounted for by torsional diffusion is the variability in *ring conformations*. Since the torsion angles at bonds inside cycles cannot be independently varied, our framework treats them as part of the local structure. Therefore, torsional diffusion relies on the local structure sampler $p_G(L)$ to accurately model cycle conformations. Although this is true for a large number of relatively small rings (especially aromatic ones) present in many drug-like molecules, it is less true for puckered rings, fused rings, and larger cycles. In particular, torsional diffusion does not address the longstanding difficulty that existing cheminformatics methods have with macrocycles—rings with 12 or more atoms that have found several applications in drug discovery [Driggers et al., 2008]. We hope, however, that the idea of restricting diffusion processes to the main sources of flexibility will motivate future work to define diffusion processes over cycles conformations combined with free torsion angles.

**Boltzmann generation** With Boltzmann generators we are typically interested in sampling the Boltzmann distribution over the entire (Euclidean) conformational space $p_G(C)$. However, the procedure detailed in Section 3.6 generates (importance-weighted) samples from the Boltzmann distribution *conditioned* on a given local structure $p_G(C \mid L)$. To importance sample from the full Boltzmann distribution $p_G(C)$, one would need a model $p_G(L)$ over local structures that also provides exact likelihoods. This is not the case with RDKit or, to the best of our knowledge, other existing models, and therefore an interesting avenue for future work.

**Proteins** As protein conformations are often described with backbone dihedral (i.e., torsion angles), it is natural to consider whether torsional diffusion may be useful for modeling protein flexibility. However, we do not believe that the *direct* application of the framework to proteins or other macro-molecules is very promising. Small changes in torsional coordinates cause large displacements in distant regions of the molecule, so the influence on a torsional score is not limited to the local neighborhood of the bond. For small molecules—even the ones in GEOM-XL—this is not a problem because of their limited spatial and graph theoretic diameters. In proteins, however, the graph diameter is 3 times the sequence length and can easily reach over 1000; and interactions between distant residues are extremely important in determining the structure and constraining flexibility. Although torsional diffusion may not be the right framework for modeling proteins, we believe that similar ideas (i.e., well-chosen diffusions over the flexible degrees of freedom) could be useful for generative models of protein structure and is a promising avenue of work.

## G  Experimental details

### G.1  Dataset details

**Splits** We follow the data processing and splits from Ganea et al. [2021]. The splits are random with train/validation/test of 243473/30433/1000 for GEOM-DRUGS and 106586/13323/1000 for GEOM-QM9. GEOM-XL consists of only a test split (since we do not train on it), which consists of all 102 molecules in the MoleculeNet dataset with at least 100 atoms. For all splits, the molecules whose CREST conformers all have a canonical SMILES different from the SMILES of the molecule (meaning a reacted conformer), or that cannot be handled by RDKit, are filtered out.

**Dataset statistics** As can be seen in Figure 7, the datasets differ significantly in molecule size as measured by number of atoms or rotatable bonds. Particularly significant is the domain shift between DRUGS and XL, which we leverage in our experiments by testing how well models trained on DRUGS generalize to XL.

**Boltzmann generator** The torsional Boltzmann generator described in Section 4.5 is trained and tested on molecules from GEOM-DRUGS with 3–7 rotatable bonds. The training (validation) set consists of 10000 (400) such randomly selected molecules from the DRUGS training (validation) set. The test set consists of all the 453 molecules present in the DRUGS test set with 3–7 rotatable bonds.

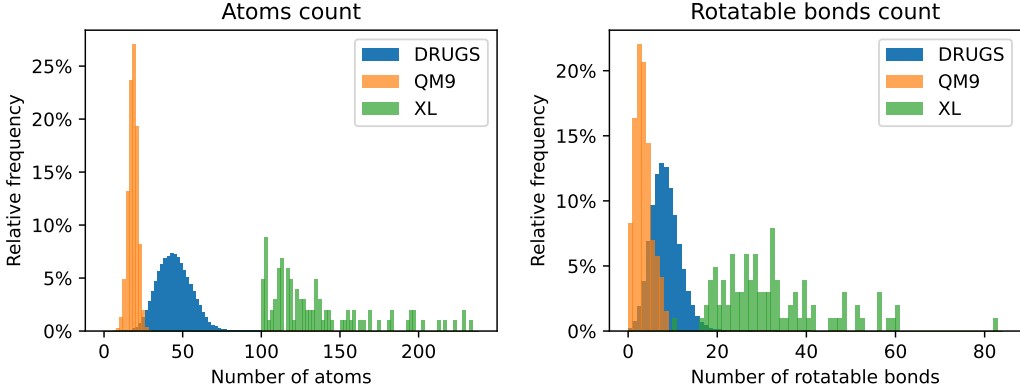

Figure 7: Statistics about the atoms and rotatable bonds counts in the three different datasets.

### G.2 Training and tuning details

**Conformer generation** For conformer ensemble generation on GEOM-DRUGS, the torsional diffusion models were trained on NVIDIA RTX A6000 GPUs for 250 epochs with the Adam optimizer (taking from 4 to 11 days on a single GPU). The hyperparameters tuned on the validation set were (in bold the value that was chosen): initial learning rate (0.0003, **0.001**, 0.003), learning rate scheduler patience (5, **20**), number of layers (2, **4**, 6), maximum representation order (1st, **2nd**), $r_{\max}$ (**5Å**, 7Å, 10Å) and batch norm (**True**, False). All the other default hyperparameters used can be found in the attached code. For GEOM-XL the same trained model was used; for GEOM-QM9 a new model with the same hyperparameters was trained.

**Torsional Boltzmann generators** We start from a torsional diffusion model pretrained on GEOM-DRUGS, and train for 250 epochs (6-9 days on a single GPU). A separate model is trained for every temperature. The resampling procedure with 5 steps is run for every molecule every $\max(5, ESS)$ epochs, where $ESS$ is computed for the current set of 32 samples. The only hyperparameter tuned (at temperature 300K) is $\sigma_{\min}$, the noise level at which to stop the reverse diffusion process.

We further improve the training procedure of torsional Boltzmann generators by implementing *annealed training*. The Boltzmann generator for some temperature $T$ is trained at epoch $k$ by using the Boltzmann distribution at temperature $T' = T + (3000 - T)/k$ as the target distribution for that epoch. Intuitively, this trains the model at the start with a smoother distribution that is easier to learn, which gradually transforms into the desired distribution.

### G.3 Evaluation details

**Ensemble RMSD** As evaluation metrics for conformer generation, Ganea et al. [2021] and following works have used the so-called Average Minimum RMSD (AMR) and Coverage (COV) for Precision (P) and Recall (R) measured when generating twice as many conformers as provided by CREST. For $K = 2L$ let $\{C_l^*\}_{l \in [1,L]}$ and $\{C_k\}_{k \in [1,K]}$ be respectively the sets of ground truth and generated conformers:

$$
\begin{aligned}
\text{COV-R} &:= \frac{1}{L} \left| \{l \in [1..L] : \exists k \in [1..K], \text{RMSD}(C_k, C_l^*) < \delta \right| \\
\text{AMR-R} &:= \frac{1}{L} \sum_{l \in [1..L]} \min_{k \in [1..K]} \text{RMSD}(C_k, C_l^*)
\end{aligned}
\tag{35}
$$

where $\delta$ is the coverage threshold. The precision metrics are obtained by swapping ground truth and generated conformers.

In the XL dataset, due to the size of the molecules, we compute the RMSDs without testing all possible symmetries of the molecules, therefore the obtained RMSDs are an upper bound, which we find to be very close in practice to the permutation-aware RSMDs.

**Runtime evaluation**   We benchmark the methods on CPU (Intel i9-9920X) to enable comparison with RDKit. The number of threads for RDKit, `numpy`, and `torch` is set to 8. We select 10 molecules at random from the GEOM-DRUGS test set and generate 8 conformers per molecule using each method. Script loading and model loading times are not included in the reported values.

**Boltzmann generator**   To evaluate how well the torsional Boltzmann generator and the AIS baselines sample from the conditional Boltzmann distribution, we report their median effective sample size (ESS) [Kish, 1965] given the importance sampling weights $w_i$ of 32 samples for each molecule:

$$ESS = \frac{\left(\sum_{i=1}^{32} w_i\right)^2}{\sum_{i=1}^{32} w_i^2} \tag{36}$$

This approximates the number of independent samples that would be needed from the target Boltzmann distribution to obtain an estimate with the same variance as the one obtained with the importance-weighted samples.

For the baseline annealed importance samplers, the transition kernel is a single Metropolis-Hastings step with the wrapped normal distributions on $\mathbb{T}^m$ as the proposal. We run with a range of kernel variances: $0.25, 0.5, 0.3, 0.5, 0.75, 1, 1.5.2$; and report the best result. We use an exponential annealing schedule; i.e., $p_n \propto p_0^{1-n/N} p_N^{n/N}$ where $p_0$ is the uniform distribution and $p_N$ is the target Boltzmann density.

# H   Additional results

**Performance vs size**   Figure 8 shows the performance of different models as a function of the number of rotatable bonds. Molecules with more rotatable bonds are more flexible and are generally larger; it is therefore expected that the RMSD error will increase with the number of bonds. With very few rotatable bonds, the error of torsional diffusion depends mostly on the quality of the local structures it was given, and therefore it has a similar error as RDKit. However, as the number of torsion angles increases, torsional diffusion deteriorates more slowly than other methods.

The trend continues with the very large molecules in GEOM-XL (average 136 atoms and 32 rotatable bonds). These not only are larger and more flexible, but—for machine learning models trained on GEOM-DRUGS—are also out of distribution. As shown in Table 6, on GEOM-XL GeoMol only performs marginally better than RDKit, while torsional diffusion reduces RDKit AMR by 30% on recall and 12% on precision. These results can very likely be improved by training and tuning the torsional diffusion model on larger molecules.

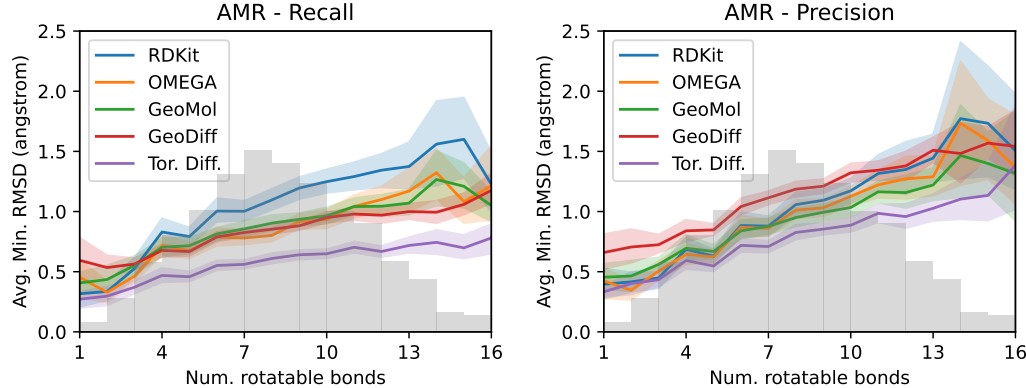

Figure 8: Average minimum RMSD (AMR) for recall (*left*) and precision (*right*) of the different conformer generation methods for molecules with different number of rotatable bonds in GEOM-DRUGS. The background shows the distribution of the number of rotatable bonds.

Table 6: Performance of various methods on the GEOM-XL dataset.

| Model | AMR-R ↓ | | AMR-P ↓ | |
|---|---|---|---|---|
| | Mean | Med | Mean | Med |
| RDKit | 2.92 | 2.62 | 3.35 | 3.15 |
| GeoMol | 2.47 | 2.39 | 3.30 | 3.15 |
| Torsional Diffusion | **2.05** | **1.86** | **2.94** | **2.78** |

Table 7: Performance of various methods on the GEOM-QM9 dataset test-set ($\delta = 0.5$Å). Again GeoDiff was retrained on the splits from Ganea et al. [2021].

| | Recall | | | | Precision | | | |
|---|---|---|---|---|---|---|---|---|
| | Coverage ↑ | | AMR ↓ | | Coverage ↑ | | AMR ↓ | |
| Method | Mean | Med | Mean | Med | Mean | Med | Mean | Med |
| RDKit | 85.1 | **100.0** | 0.235 | 0.199 | 86.8 | **100.0** | 0.232 | 0.205 |
| OMEGA | 85.5 | **100.0** | **0.177** | **0.126** | 82.9 | **100.0** | 0.224 | **0.186** |
| GeoMol | 91.5 | **100.0** | 0.225 | 0.193 | 86.7 | **100.0** | 0.270 | 0.241 |
| GeoDiff | 76.5 | **100.0** | 0.297 | 0.229 | 50.0 | 33.5 | 0.524 | 0.510 |
| Torsional diffusion | **92.8** | **100.0** | 0.178 | 0.147 | **92.7** | **100.0** | **0.221** | 0.195 |

**Small molecules** We also train and evaluate our model on the small molecules from GEOM-QM9 and report the performance in Table 7. For these smaller molecules, cheminformatics methods already do very well and, given the very little flexibility and few rotatable bonds present, the accuracy of local structure significantly impacts the performance of torsional diffusion. RDKit achieves a mean recall AMR just over 0.23Å, while torsional diffusion based on RDKit local structures results in a mean recall AMR of 0.178Å. This is already very close lower bound of 0.17Å that can be achieved with RDKit local structures (as approximately calculated by conformer matching). Torsional diffusion does significantly better than other ML methods, but is only on par with or slightly worse than OMEGA, which, evidently, has a better local structures for these small molecules.

**Ablation experiments** In Table 8 we present a set of ablation studies to evaluate the importance of different components of the proposed torsional diffusion method:

1. *Baseline* refers to the model described and tested throughout the paper.

2. *Probability flow ODE* refers to using the ODE formulation of the reverse diffusion process (not an ablation, strictly speaking). As expected, it obtains similar results to the baseline SDE formulation.

3. *Only D.E. matching* refers to a model trained on conformers obtained by a random assignment of RDKit local structures to ground truth conformers (without first doing an optimal assignment as in Appendix E); this performs only marginally worse than full conformer matching.

4. *First order irreps* refers to the same model but with node irreducible representations kept only until order $\ell = 1$ instead of $\ell = 2$; this worsens the average error by about 5%, but results in a 41% runtime speed-up.

5. *Train on ground truth L* refers to a model trained directly on the ground truth conformers without conformer matching but tested (as always) on RDKit local structures; although the training and validation score matching loss of this model is significantly lower, its inference performance reflects the detrimental effect of the local structure distributional shift.

6. *No parity equivariance* refers to a model whose outputs are parity invariant instead of parity equivariant; the model cannot distinguish a molecule from its mirror image and fails to learn, resulting in performance on par with a random baseline.

7. *Random $\tau$* refers to a random baseline using RDKit local structures and uniformly random torsion angles.

Table 8: Ablation studies with ensemble RMSD on GEOM-DRUGS. Refer to the text in the Appendix for an explanation of each entry. As usual, we compute Coverage with $\delta = 0.75$ Å.

| | Recall | | | | Precision | | | |
|---|---|---|---|---|---|---|---|---|
| | Coverage ↑ | | AMR ↓ | | Coverage ↑ | | AMR ↓ | |
| Method | Mean | Med | Mean | Med | Mean | Med | Mean | Med |
| Baseline | 72.7 | 80.0 | 0.582 | 0.565 | 55.2 | **56.9** | **0.778** | **0.729** |
| Probability flow ODE | **73.1** | 80.4 | **0.577** | **0.557** | **55.3** | 55.7 | 0.779 | 0.737 |
| Only D.E. matching | 72.5 | **81.1** | 0.588 | 0.569 | 53.8 | 56.1 | 0.794 | 0.749 |
| First order irreps | 70.1 | 77.9 | 0.605 | 0.589 | 51.4 | 51.4 | 0.817 | 0.783 |
| Train on ground truth $L$ | 34.8 | 22.4 | 0.920 | 0.909 | 22.3 | 7.8 | 1.182 | 1.136 |
| No parity equivariance | 30.5 | 12.5 | 0.928 | 0.929 | 17.9 | 3.9 | 1.234 | 1.217 |
| Random $\tau$ | 30.9 | 13.2 | 0.922 | 0.923 | 18.2 | 4.0 | 1.228 | 1.217 |

Table 9: Ensemble RMSD results on GEOM-DRUGS for varying number of diffusion steps. 20 steps were used for all results reported elsewhere. As usual, we compute Coverage with $\delta = 0.75$ Å.

| | Recall | | | | Precision | | | |
|---|---|---|---|---|---|---|---|---|
| | Coverage ↑ | | AMR ↓ | | Coverage ↑ | | AMR ↓ | |
| Steps | Mean | Med | Mean | Med | Mean | Med | Mean | Med |
| 3 | 42.9 | 33.8 | 0.820 | 0.821 | 24.1 | 11.1 | 1.116 | 1.100 |
| 5 | 58.9 | 63.6 | 0.698 | 0.685 | 35.8 | 26.6 | 0.979 | 0.963 |
| 10 | 70.6 | 78.8 | 0.600 | 0.580 | 50.2 | 48.3 | 0.827 | 0.791 |
| 20 | 72.7 | 80.0 | 0.582 | 0.565 | 55.2 | 56.9 | 0.778 | 0.729 |
| 50 | 73.1 | 80.4 | 0.578 | 0.557 | 57.6 | 60.7 | 0.753 | 0.699 |

Table 10: Median absolute error of generated v.s. ground truth ensemble properties with and without relaxation. $E, \Delta\epsilon, E_{\min}$ in kcal/mol, $\mu$ in debye.

| | Without relaxation | | | | With relaxation | | | |
|---|---|---|---|---|---|---|---|---|
| Method | $E$ | $\mu$ | $\Delta\epsilon$ | $E_{\min}$ | $E$ | $\mu$ | $\Delta\epsilon$ | $E_{\min}$ |
| RDKit | 39.08 | 1.40 | 5.04 | 39.14 | 0.81 | 0.52 | 0.75 | 1.16 |
| OMEGA | **16.47** | **0.78** | **3.25** | **16.45** | 0.68 | 0.66 | 0.68 | 0.69 |
| GeoMol | 43.27 | 1.22 | 7.36 | 43.68 | 0.42 | **0.34** | 0.59 | 0.40 |
| GeoDiff | 18.82 | 1.34 | 4.96 | 19.43 | 0.31 | 0.35 | 0.89 | 0.39 |
| Tor. Diff. | 36.91 | 0.92 | 4.93 | 36.94 | **0.22** | 0.35 | **0.54** | **0.13** |

**Reverse diffusion steps**   In Table 9 we vary the number of steps used in the reverse diffusion process and evaluate the ensemble RMSD results on GEOM-DRUGS. We find that torsional diffusion is remarkably parsimonious in terms of number of steps required: the majority of gain in performance over prior diffusion-based methods is attained with only 10 steps. We confirm that increasing the number of steps from the default of 20 to 50 only results in minor performance gains.

**Ensemble properties**   In Table 10, we report the median absolute errors of the Boltzmann-weighted properties of the generated vs CREST ensembles, with and without GFN2-xTB relaxation. For all methods, the errors without relaxation are far too large for the computed properties to be chemically useful—for reference, the thermal energy at room temperature is 0.59 kcal/mol. In realistic settings, relaxation of local structures is necessary for any method, after which errors from global flexibility become important. After relaxation, torsional diffusion obtains property approximations on par or better than all previous methods.

**Torsional Boltzmann generator**   Figure 9 shows the histograms of ESSs at 500K for the torsional Boltzmann generator and the AIS baseline. While AIS fails to generate more than one effective sample for most molecules (tall leftmost column), torsional Boltzmann generators are much more efficient, with more than five effective samples for a significant fraction of molecules.

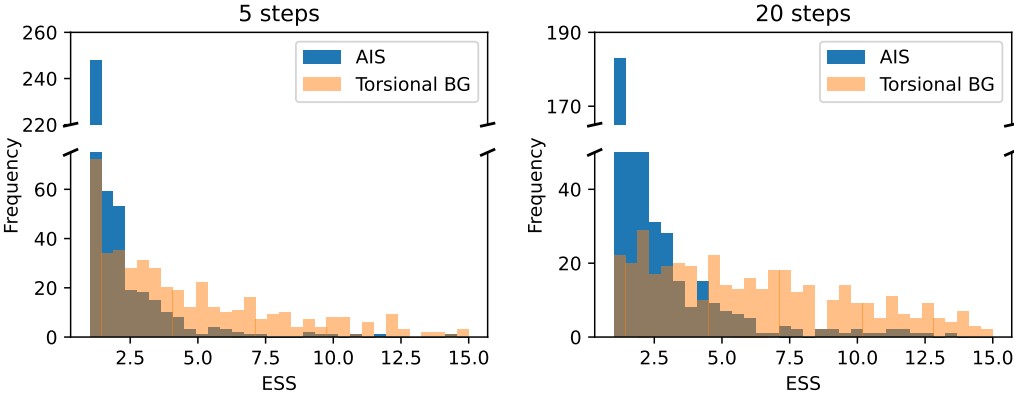

Figure 9: Histogram of the ESSs of the torsional Boltzmann generator and AIS baseline at 500K.