# OpenReview forum: "Torsional Diffusion for Molecular Conformer Generation"
_NeurIPS.cc/2022/Conference — NeurIPS 2022 Accept_

### Official Review · Reviewer_P55i · 2022-07-06

**Rating:** 7
**Confidence:** 4
**Soundness:** 3 good
**Presentation:** 4 excellent
**Contribution:** 3 good

**Summary:**

This paper argues that instead of generating bond length, angle and torsion simultaneously, one should focus on the more difficult part, i.e. the torsion. And the authors further propose a diffusion model working on this Riemannian manifold of the torsion angles. A Bolzman generator is then introduced based on the likelihood calculated from the diffusion model. Experiments on molecules with normal size, large size and small size datasets show increased performance in terms of RMSD and convergence time.


**Questions:**


Please see the above weaknesses.

**Limitations:**

The authors have adequately discussed the limitations of torsional diffusion, which I think is quite fair.

**Strengths And Weaknesses:**

Pros:
-The paper is well motivated to work on the torsion angles only, which reasonably leads to advantages in speed and performance, under mild assumptions about the bond length and angle. The idea is novel.
-The idea of using the diffusion model on the torsion-composed manifold is compatible with the Bolzman generator and leads to some good properties.
-The paper is well written. The idea, the procedure, why symmetry properties are satisfied, how to solve the sample-wise intrinsic coordinates, are well elaborated.

Cons:
-Ablation studies, e.g. about the depth of diffusion, conformed matching and any other hyperparameters would help to understand and are desired.

---

> ### Author Response · Authors · 2022-07-31
> **Response to Reviewer P55i**
>
> We thank the reviewer for the useful feedback. We are glad that the reviewer found the idea "novel" and the paper "well-written". We agree with the reviewer on the importance of ablation studies and report some additional results below. We will integrate these into the paper along with existing ablation studies in Appendix H.2.
>
> **Depth of diffusion.** We report additional results (all medians) on how the model's performance varies with a different number of inference steps (same trained model; in the paper we report numbers for 20 steps). As expected, with more steps the performance increases, but with only 10 steps the performance is already significantly better than any previous method.
>
> ||Cov-R|AMR-R|Cov-P|AMR-P|
> |-|-|-|-|-|
> |3 steps|33.8%|0.821|11.1%|1.100|
> |5 steps|63.6%|0.685|26.6%|0.963|
> |10 steps|78.8%|0.580|48.3%|0.791|
> |20 steps|80.0%|0.565|56.9%|0.729|
> |50 steps|80.4%|0.557|60.7%|0.699|
> |OMEGA|54.6%|0.762|33.3%|0.854|
>
>
>
> **Other ablations.** We also complement the ablations already present in appendix H.2 (reported again below) with some new ones (all medians).
>
> ||Cov-R|AMR-R|Cov-P|AMR-P|
> |-|-|-|-|-|
> |Baseline|80.0%|0.565|56.9%|0.729|
> |First order irreps|77.9%|0.589|51.4%|0.783|
> |Only D.E. matching|81.8%|0.569|56.1%|0.749|
> |Train on ground truth L|22.4%|0.909|7.8%|1.136|
> |No parity equivariance|12.5%|0.929|3.9%|1.217|
> |Probability flow ODE|57.7%|0.557|55.7%|0.737|
> |Random torsion|13.2%|0.923|4.0%|1.217|
>
>
> 1. _Baseline_ refers to the model described and tested throughout the paper.
> 2. _First-order irreps_ refers to the same model but with node irreducible representations kept only until order $\ell = 1$ instead of $\ell = 2$; this worsens the average error by about 5%, but results in a 41% runtime speed-up.
> 3. _Only D.E. matching_ refers to a model trained on conformers obtained by a random assignment of RDKit local structures to ground truth conformers (without first doing an optimal assignment as in Appendix E); this performs only marginally worse than full conformer matching.
> 4. _Train on ground truth L_ refers to a model trained directly on the ground truth conformers without conformer matching but tested (as always) on RDKit local structures; although the training and validation score matching loss of this model is significantly lower, its inference performance reflects the detrimental effect of the local structure distributional shift.
> 5. _No parity equivariance_ refers to a model whose outputs are parity invariant instead of parity equivariant; the model cannot distinguish a molecule from its mirror image and fails to learn, resulting in performance on par with a random baseline.
> 6. _Probability flow ODE_ refers to using the ODE formulation of the reverse diffusion process as described in Song et al. [1]. As expected it obtains analogous results to the baseline SDE formulation.
> 7. _Random torsion_ refers to a random baseline using RDKit local structures and uniformly random torsion angles.
>
> We thank the reviewer again for the actionable feedback. Please let us know if there is any other ablation you could find helpful. We hope that these experiments address the reviewer's concerns and increase their confidence in the quality of our work.
>
>
> [1] Song, Yang, et al. "Score-based generative modeling through stochastic differential equations." ICLR 2021.

---

### Official Review · Reviewer_hcv8 · 2022-07-08

**Rating:** 7
**Confidence:** 5
**Soundness:** 3 good
**Presentation:** 3 good
**Contribution:** 3 good

**Summary:**

This paper studies molecular conformation generation. Different from previous works that focus on coordinate prediction or distance prediction, this paper proposes to focus solely on torsional angles of rotatable bonds in molecules, leaving all other degrees of freedom fixed.

Inspired by diffusion models on Riemannian manifolds, the authors first lay the theoretical framework of torsional diffusion on hypertorus following the standard denoising diffusion models. Based on this framework, they show that such a diffusion process can also be defined on Cartesian coordinates, and design a SE(3)-invariant and parity equivariant score network operating on 3D point clouds.

They show the proposed method achieves state-of-the-art performance on GEOM-DRUGS, with much fewer denoising steps compared to previous models. The resulting method is also capable of exact likelihoods estimation, making it suitable for matching the Boltzmann distribution over torsion angles using the energy function.

**Questions:**

In line 111, the authors give the definition of the freely rotatable bond, which are those that serve as a bridge between two components. Is this definition the same as the definition provided by RDKit API? (As I remember we can get all rotatable bonds of a molecule by calling a certain function provided by RDKit).

As far as I understand, torsion angles in cycles are also the degrees of freedom that determine the conformation of a molecule. How are these rotatable bonds handled in the proposed method? Can we also do diffusion on these bonds?

**Limitations:**

The authors didn't discuss limitations and potential negative societal impact of their work.

**Strengths And Weaknesses:**

### Strengths
Torsional angles are the most important degrees of freedom that determine the conformations of a molecule. Focusing on torsional angle modeling for conformation prediction is a straightforward and novel idea. To the best of my knowledge, it is the first work that marries denoising diffusion models with torsional angle prediction, and I like this idea.

The resulting score functions for torsional angle diffusion is SE(3)-invariant and parity equivariant for every rotatable bond in a molecule, which is a nice property.

The proposed model is capable of exact likelihood estimation, which is another nice property thanks to the use of SDE.

### Weaknesses
The model relies on an external algorithm, such as RDKit ETKDG, to generate the initial local structures for the target molecule, and the only degrees of freedom are torsional angles. This means the chirality of the conformation is entirely determined by the initial structure, or equivalently, by the external algorithm that provides the initial guess structure.

---

> ### Author Response · Authors · 2022-07-31
> **Response to Reviewer hcv8**
>
> We thank the reviewer for the thorough review and positive feedback. We respond to the reviewer’s questions and concerns below, and refer to appendix F where some of the points have been discussed. In the revision, we will update the appendix and better link it to the main text.
>
> **"The model relies on an external algorithm, such as RDKit ETKDG, to generate the initial local structures"**
>
> This is true, but we view it as a strength rather than a weakness. Generating local structures is a well-studied and mostly solved problem in computational chemistry. In Appendix F.1, for example, we verify the high accuracy of local structures provided by RDKit. By leveraging strong existing methods, we simplify the learning task and ultimately obtain superior results. Although one day the accuracy of these external algorithms may be limiting, at present the benefits of leveraging them far outweigh the downsides of not learning local structures.
>
> **"chirality of the conformation is entirely determined by the initial structure"**
>
> This is also correct. However, this is actually desirable because, since chiralities cannot interconvert under normal conditions, the desired chirality (at each chiral center) is considered a property of the molecule and is therefore specified as input. Indeed, in the GEOM dataset [1] conformers with different chiralities belong to different conformer ensembles. Since we get the local structures from RDKit, which has been implemented to correctly set the chirality, we are guaranteed to have the right chirality in the final molecule. This is a significant advantage over previous ML models like GeoDiff [2] where the user has no way to impose the right chirality. For a more comprehensive discussion on chemical isomerisms we refer to appendix F.3.
>
> **"Is the rotatable bond definition the same as the definition provided by RDKit API?"**
>
> Almost but not exactly. RDKit has the same definition with the notable difference that it does not consider double bonds as rotatable. Whether to include or not double bonds is a design choice and torsional diffusion works either way. We included them in the diffused components because although they are more physically constrained than other types of freely rotatable bonds they still have some flexibility that we wanted to model.
>
> **"How are rotatable bonds in rings handled in the proposed method?"**
>
> At present, torsion angles inside rings are considered part of the local structure and are therefore determined by RDKit. Empirically, these are well-determined, as verified along with other aspects of local structure in Appendix F.1. Chemically, this is because most rings in our dataset are small. However, in cases where ring flexibility is important, such as macrocycles (appendix F.4), it would be more challenging to improve over RDKit. Extending torsional diffusion to such cases is not straightforward since the torsion angles there are interdependent with each other and with bond angles. However, we believe that future work could use the same principles we developed to add the right degrees of freedom to model such flexibility. Possible ideas to do so can be found in the chemistry literature: breaking one bond per cycle to make the torsions freely rotatable [3] or using puckering coordinates to model cycles conformation on a hypersphere [4].
>
> **"limitations and potential negative societal impact of their work."**
>
> We refer to appendix F.4 (and the responses to previous questions) for some discussion of the limitations of our work. The potential societal impact of this work lies mainly in the fields and applications where molecular conformer generation is already used; i.e., many chemistry and chemical engineering domains. These vary widely in their impact on society and have historically been associated with both positive and negative developments. However, in light of the essential role of chemistry in both healthcare and renewable energy, we believe our work is more likely to lead to positive rather than negative societal impacts.
>
> We thank the reviewer again for the actionable feedback, we will integrate these discussions with those in appendix F and better reference them in the main text. We hope our responses address the reviewer's concerns and increase their confidence in the quality of our work.
>
> [1] Axelrod, Simon, and Rafael Gomez-Bombarelli. "GEOM, energy-annotated molecular conformations for property prediction and molecular generation." Scientific Data 9.1 (2022)
>
> [2] Xu, Minkai, et al. "GeoDiff: A geometric diffusion model for molecular conformation generation.", ICLR 2022
>
> [3] Abagyan, Ruben A., and Alexey K. Mazur. "New Methodology for Computer-Aided Modelling of Biomolecular Structure and Dynamics 2. Local Deformations and Cycles." Journal of Biomolecular Structure and Dynamics 6.4 (1989): 833-845.
>
> [4] Cremer, D. T., and J. A. Pople. "General definition of ring puckering coordinates." Journal of the American Chemical Society (1975)

---

### Official Review · Reviewer_7ogf · 2022-07-11

**Rating:** 7
**Confidence:** 3
**Soundness:** 4 excellent
**Presentation:** 3 good
**Contribution:** 3 good

**Summary:**

This paper propose torsional diffusion for molecular conformer generation. Different from the previous conformer generation work which mainly predict the position of each atoms in the 3D space, this work view the conformer generation in a different view by generating the torsion. To achieve this, the paper also provides the exact likelihood of the generated conformers, which lead a better training pipeline comparing with the previous work. From the experiment side, the paper reaches a very good SOTA result comparing with GEOM-DRUGS dataset, which is very convincing.

In addition, the paper propose torsional Boltzmann generator that can be expand and adapt in generating various class of molecules.

**Questions:**

For GeoMol and GeoDiff's performance, it seems have a huge gap between the original paper. I am not sure if it is reasonable since the author uses a different data split. It may be fair to use their split for another comparison.

**Limitations:**

Same as the Questions Section.

**Strengths And Weaknesses:**

The paper is in a good writing, very easy to follow and the idea is natural. Rather than learning the Euclidian information in the previous task, learning the torsion to generate conformer is more reasonable in a physical natural. Although learning torsion was proposed in GeoMol using MPNN, enhancing it use Diffusion model is still soundable. The experiment result is solid with many details.
For the weakness, I know from the author's statement, they retrain the GeoDiff to adapt the larger dataset, but it still seems the GEOM-DRUGS and GEOM-QM9's performance for the baseline GeoDiff and GeoMol is much lower than their report number. Any suggestion towards this huge gap? Is it possible to use their split to have another comparison?

---

> ### Author Response · Authors · 2022-07-31
> **Response to Reviewer 7ogf**
>
> We thank the reviewer for the positive review. We are glad that the reviewer found the paper "easy to follow" and the idea "natural". The reviewer raises concerns about a gap in the performance of the baselines GeoMol and GeoDiff relative to their original papers. Below, we demonstrate that there is in fact no gap and we try to clarify possible sources of misunderstanding.
>
> Regarding **GeoMol**, we use the splits, evaluation code, and pretrained models directly from the GeoMol repository [1]. We generate conformers on the test set using the provided inference script and obtain results **very similar** to those reported in the original paper [2]. (See table below).
>
> Regarding **GeoDiff**, in order to compare with GeoMol and the other baselines, we do indeed retrain the model on the GeoMol splits (which unlike the ones in GeoDiff do not include filters on the number of conformers per molecule). We do so using exactly the same code and hyperparameters provided by the authors [3], and obtain results on the GeoMol test set that are **very similar to** (and even slightly better than) those reported on the GeoDiff test set in the original paper [4]. (See table below).
>
> The **only difference** between our main Table 1 and the tables in the GeoDiff and GeoMol papers is that we report Coverage metrics with a **threshold of $\delta=0.75$ angstroms instead of $\delta=1.25$**. The mathematical expression for Coverage as a function of $\delta$ is shown in Appendix G.3, eq 35. In brief, a lower $\delta$ means that generated and true conformers have to be more similar in order to be considered the same, so the coverage will naturally be lower for lower $\delta$.
>
> Previous papers used $\delta=1.25$ since it was effective at differentiating existing methods at the time. However, GeoMol, GeoDiff, and our method all easily achieve 95% median recall on GEOM-Drugs with $\delta=1.25$. Given space constraints in the main table, we elected to show only Coverage with a more stringent threshold $\delta=0.75$.
>
> **Complete results for the full range of $\delta$** are shown in Appendix H.1 (Our model is always the top-performing method regardless of $\delta$). In particular, in the table below we also show Coverage with $\delta=1.25$ to show that the ensemble quality we obtain with GeoMol and GeoDiff are indeed consistent with those originally reported.
>
> **Results in our Table 1 and Appendix H.1** (all medians)
> ||Cov-R, $\delta=1.25$|Cov-R, $\delta=0.75$|AMR-R|Cov-P, $\delta=1.25$|Cov-P, $\delta=0.75$|AMR-P|
> | ----------- | :-----------: | :-----------: | :-----: | :-----: | :----: | :---: |
> | GeoMol      | 93.9%       | 41.4% | 0.834 | 95.6% | 36.4% | 0.841 |
> | GeoDiff   | 100%        | 37.8% |  0.809 | 69.5% | 14.5% | 1.090 |
>
>
>
> **Results from the original GeoMol and GeoDiff papers** (Table 1 in both) (all medians)
>
> |      | Cov-R, $\delta=1.25$ | Cov-R, $\delta=0.75$ | AMR-R | Cov-P, $\delta=1.25$ | Cov-P, $\delta=0.75$| AMR-P |
> | ----------- | :-----------: | :-----------: | :-----: | :-----: | :----: | :---: |
> | GeoMol      | 95.1%       | --- | 0.837 | 94.4% |---  | 0.856 |
> | GeoDiff   | 97.9%        | ---|  0.853 | 64.6% |--- | 1.123 |
>
> We thank the reviewer for raising this clarity issue and will explain these results more thoroughly in the revision. We hope these points resolve the reviewer's concern and increase the reviewer's confidence in the quality of our work.
>
> [1] https://github.com/PattanaikL/GeoMol
>
> [2] Ganea, Octavian, et al. "GeoMol: Torsional geometric generation of molecular 3d conformer ensembles.", NeurIPS 2021
>
> [3] https://github.com/MinkaiXu/GeoDiff
>
> [4] Xu, Minkai, et al. "GeoDiff: A geometric diffusion model for molecular conformation generation.", ICLR 2022

---

> > ### Comment · Reviewer_7ogf · 2022-08-08
> > **Response to Author**
> >
> > Thanks for the reply, it clears most of my concern. I also take other reviewers opinion as input, since all of the reviewers give positive feed back and I am not super confidence in this area, I will keep my current score.

---

### Official Review · Reviewer_QmNA · 2022-07-11

**Rating:** 8
**Confidence:** 3
**Soundness:** 4 excellent
**Presentation:** 3 good
**Contribution:** 4 excellent

**Summary:**

This paper proposed a novel diffusion model for molecular conformer generation. The proposed diffusion model is operated in the space of torsion angles of the conformer. An extrinsic-to-intrinsic score model is learned for the diffusion process to predict the torsional scores directly from the 3D point cloud representation of the conformer. The exact likelihoods of the generated conformers can be computed to enable energy-based training with samples from the Boltzmann distribution. The proposed diffusion model outperforms existing machine learning and cheminformatic-based solutions on the GEOM benchmark.

**Questions:**

I am very excited to see that the proposed method outperforms commercial software for molecular conformation prediction. I am interested in the potential of such a model in the future. What are the main practical limitations (e.g., sample efficiency, training time, inference time, ...) for deploying and using such a model in an industrial scenario? Can it be considered for high-throughput applications? What is the main challenge of adapting the model to larger molecules (e.g., proteins) use cases?

**Limitations:**

The authors have adequately addressed the limitations in the appendix.

**Strengths And Weaknesses:**

This paper proposed a novel diffusion model that has a strong empirical impact on the fundamental task of molecular conformer generation. With comprehensive experiments, the authors showed that the model is able to generate more accurate conformers in less time compared with existing machine learning solutions. The model also consistently outperforms a commercial software OMEGA on the GEOM benchmark. The evaluations were conducted with high quality, with sufficient details, strong quantitative evidence, and adequate ablation analysis.

Theoretically, the intuitions behind the important design decisions are well explained, i.e., learning a diffusion model in the torsional space instead of euclidean space, using a SE(3)-invariant model to parameterize the score function, etc. The theory behind the diffusion model and training algorithm is also sound.

---

> ### Author Response · Authors · 2022-07-31
> **Response to Reviewer QmNA part 1**
>
> We thank the reviewer for the positive review of the paper and appendix and for recognizing the "sound theory" and the "strong empirical impact" of our work. We are also very excited about the potential of the method, and have been in discussions with industrial collaborators to maximize its practical impact. These discussions have included many of the factors pointed out by the reviewer.
>
> **“What are the main practical limitations (e.g., sample efficiency, training time, inference time, ...) for deploying and using such a model in an industrial scenario?”**
>
> **Sample efficiency**
> * Unlike other ML approaches, torsional diffusion substantially reduces the data-space dimensionality (from 132 to 8 in the GEOM dataset on average) and builds on existing chemical knowledge. Thus, we actually anticipate sample efficiency to be a strength of the method.
> * Our method is particularly efficient for sampling the Boltzmann distribution. Unlike previous Boltzmann generators [1,2], ours is generalizable (or transferable) across the chemical space, which means that it can be applied directly to new molecules in high-throughput workflows.
>
> **Training time**
> * Because of the size and breadth of the training set, our trained model should be suitable out-of-the-box for most applications. We envision our method as playing a similar role in workflows as RDKit ETKDG or OMEGA, which are distributed with a universal set of parameters.
> * Should practitioners wish to retrain or fine-tune on closed-source data or a different distribution, our model is lightweight and requires only modest resources — a single GPU for 4-11 days to convergence (appendix G.2), with the majority of performance attained in significantly less time.
>
> **Inference runtime**
> * The best evaluated model is admittedly slower than the fastest cheminformatics and ML methods. However, it can be accelerated by reducing the number of inference steps and using first-order representations with a minor loss in performance (table 2 and appendix H.2). Additionally, concrete steps exist to accelerate the codebase, which currently has not been optimized for runtime.
> * The current runtimes (approx. 10 core-mins per ensemble of 100 confs) are nonetheless orders of magnitude faster than metadynamics (approx. 90 core-hrs per ensemble with xTB [3]), already enabling the screening of millions of molecules in an industrial setting.
>
> In summary, to the best of our current knowledge, we foresee no practical _methodological_ limitations to deploying the method as a substitute for RDKit ETKDG or OMEGA. On the other hand, the adoption of machine learning methods in applied industries is commonly limited by **software visibility, accessibility, and engineering** considerations. Therefore, on top of open sourcing our code and models with a permissive license, we are in active discussions with the maintainers of well-established chemical libraries to directly integrate our model into their distributions. This would make it much easier and more likely for industrial practitioners to try and deploy our method in their workflows.

---

> > ### Author Response · Authors · 2022-07-31
> > **Response to Reviewer QmNA part 2**
> >
> > **“Can it be considered for high-throughput applications?”**
> >
> > We are very excited about the potential uses of our model in high-throughput industrial screening applications. In particular, two applications that we have been exploring are docking for drug discovery and property prediction for chemical design. In **drug discovery**, docking algorithms often work by evaluating the geometric complementarity of the conformer with the protein [4]. Thus, one is interested in having a high geometric recall of the conformations, so the results in Table 1 are particularly promising. On the other hand, for **chemical design**, properties of interest are largely determined by the most populated (i.e., lowest-energy) conformers. Fast conformer generation methods are used to seed optimizations with higher levels of theory in search of the global minimum. In this regard, industry researchers we spoke to were particularly impressed by the low energy gaps obtained by our relaxed structures (Table 3).
> >
> > **“What is the main challenge of adapting the model to larger molecules (e.g., proteins) use cases?”**
> >
> > Although in appendix H.1 we show improvements over other methods when extrapolating to larger molecules, we do not believe that the direct application of torsional diffusion to **model protein structure** is very promising. Small changes in torsional coordinates cause large displacements in distant regions of the molecule, so the influence on a torsional score is not limited to the local neighborhood of the bond. For small molecules—even the ones in GEOM-XL—this is not a problem because of their limited spatial and graph theoretic diameters. In proteins, however, the graph diameter is 3 times the sequence length and can easily reach over 1000; and interactions between distant residues are extremely important in determining the structure and constraining flexibility. Although torsional diffusion may not be the right framework for modeling proteins, we believe that well-chosen diffusions could be useful for generative models of protein structure and is a promising avenue of work.
> >
> > We thank the reviewer again for their review and we will incorporate some of these points of discussion in the revision. We hope this response addresses the reviewer's questions and increases their confidence in the impact of this work.
> >
> > [1] Noé, Frank, et al. "Boltzmann generators: Sampling equilibrium states of many-body systems with deep learning." Science (2019)
> >
> > [2] Köhler, Jonas, Andreas Krämer, and Frank Noé. "Smooth normalizing flows." Advances in Neural Information Processing Systems 34 (2021)
> >
> > [3] Axelrod, Simon, and Rafael Gomez-Bombarelli. "GEOM, energy-annotated molecular conformations for property prediction and molecular generation." Scientific Data 9.1 (2022)
> >
> > [4] O. Trott, A. J. Olson, AutoDock Vina: improving the speed and accuracy of docking with a new scoring function, efficient optimization and multithreading, Journal of Computational Chemistry 31 (2010)

---

> > > ### Comment · Reviewer_QmNA · 2022-08-08
> > > **Thank you for your response**
> > >
> > > Thank you for addressing all my questions! I learned a lot from your comments. Hoping to see it deployed in the industry one day!

---

### Meta-Review · Area_Chair_nR5T · 2022-08-27

**Recommendation:** Accept
**Confidence:** Certain

**Metareview:**

This paper proposes a diffusion model for molecular conformation prediction in the space of torsion angles. The idea is new, and the experimental results are strong. All reviewers like this paper.

**Award:**

No

---

### Decision · Program_Chairs · 2022-09-14

Accept